# SELECTIVE LABEL ENHANCEMENT LEARNING FOR TEST-TIME ADAPTATION

**Yihao Hu**[1,2,*]**, Congyu Qiao**[1,2,*]**, Xin Geng**[1,2]**, Ning Xu**[1,2,†]

[1] School of Computer Science and Engineering, Southeast University, Nanjing, China
[2] Key Laboratory of New Generation Artificial Intelligence Technology and
   Its Interdisciplinary Applications (Southeast University), Ministry of Education, China
`{yhhu,qiaocy,xgeng,xning}@seu.edu.cn`

## ABSTRACT

Test-time adaptation (TTA) aims to adapt a pre-trained model to the target domain using only unlabeled test samples. Most existing TTA approaches rely on definite pseudo-labels, inevitably introducing false labels and failing to capture uncertainty for each test sample. This prevents pseudo-labels from being flexibly refined as the model adapts during training, limiting their potential for performance improvement. To address this, we propose the Progressive Adaptation with Selective Label Enhancement (PASLE) framework. Instead of definite labels, PASLE assigns candidate pseudo-label sets to uncertain ones via selective label enhancement. Specifically, PASLE partitions data into confident/uncertain subsets, assigning one-hot labels to confident samples and candidate sets to uncertain ones. The model progressively trains on certain/uncertain pseudo-labeled data while dynamically refining uncertain pseudo-labels, leveraging increasing target adaptation monitored throughout training. Experiments on various benchmark datasets validate the effectiveness of the proposed approach. The source code is available at `https://github.com/palm-ml/PASLE`.

## 1 INTRODUCTION

Deep neural networks often experience performance degradation when the training and testing data are drawn from different distributions. Test-time adaptation (TTA), an emerging paradigm that adapts a pre-trained model to a different domain using only unlabeled data during testing, aims to alleviate this problem. As the practicality of TTA, it has been successfully applied across various fields, including autonomous driving (Wang et al., 2022; Volpi et al., 2022), medical image segmentation (He et al., 2021; Karani et al., 2021), and speech processing (Kim et al., 2022; Lin et al., 2022).

Most TTA approaches utilize pseudo-labeling-based methods, which first assign pseudo-labels to the unlabeled target data and then adapt the pre-trained model to the target domain through training on those pseudo-labeled samples. Some approaches utilize non-parametric classifiers to generate pseudo-labels by measuring the distance between samples and prototypical representations (Iwasawa & Matsuo, 2021) or leveraging neighboring feature similarities without model adaptation (Zhang et al., 2023). Other methods rely on the consistency between prototype-based pseudo-labels or nearest-neighbor-based pseudo-labels to guide model updates during adaptation (Jang et al., 2023; Wang et al., 2023; Sun et al., 2024). These approaches aim to improve the robustness and performance of TTA by directly using definite pseudo-labels.

Due to distribution shifts, most previous approaches inevitably introduce false pseudo-labels by adopting definite pseudo-labels. These approaches may experience significant performance deterioration. The definite pseudo-labels lose the uncertainty information of each class corresponding to the test sample, leading to the fact that the unreliable pseudo-labels cannot be flexibly tuned according to the model's increasing adaptation during the learning process. Therefore, despite making

---

[*]Equal contribution.
[†]Corresponding author.

efforts in pseudo-labeling, including prototype-based or nearest-neighbor-based methods, previous approaches usually do not fully unleash the potential of pseudo-labels for improving the performance of TTA.

To address this issue, instead of assigning definite pseudo-labels to test samples, we propose a novel framework named PASLE, i.e., Progressive Adaptation with Selective Label Enhancement. PASLE selectively assigns one-hot pseudo-labels to confident test samples or assigns candidate pseudo-label sets to uncertain test samples in a label enhancement process (Xu et al., 2019; 2020). Specifically, PASLE partitions test samples into confident and uncertain subsets based on the selective label enhancement strategy, where confident samples receive one-hot pseudo-labels, and uncertain samples are assigned candidate pseudo-label sets. The model is then iteratively trained on certain pseudo-labeled target data and uncertain pseudo-labeled target data in a progressive manner while dynamically refining the candidate pseudo-label sets of uncertain samples by exploiting the model's evolving adaptation capability to the target domain, which is monitored during the training process. Our contributions can be summarized as follows:

- We propose a selective label enhancement strategy, where PASLE partitions test samples into confident and uncertain subsets based on the model's predictive confidence scores, with confident samples receiving one-hot pseudo-labels and uncertain samples being assigned candidate pseudo-label sets.

- We introduce a progressive learning framework that trains the model on certain pseudo-labeled target data and uncertain pseudo-labeled target data in a progressive manner while dynamically refining the candidate pseudo-label sets of uncertain samples by exploiting the model's evolving adaptation capability to the target domain, which is monitored during the training process.

- We theoretically establish a generalization bound for TTA and assess the effectiveness of pseudo-labels by quantifying them through the pseudo-label error term.

## 2 RELATED WORK

**Test-time adaptation.** Domain shifts often cause machine learning systems to suffer substantial drops in performance. To address this challenge, numerous techniques have been developed to enhance model robustness against distribution shifts. Domain generalization (Zhou et al., 2022) attempts to train a model on data from one or more source domains, enabling it to generalize effectively to unseen target domains. Domain adaptation (Kouw & Loog, 2021) relies on transductive learning, where knowledge from a labeled source domain is transferred to an unlabeled target domain. Test-time adaptation (Liang et al., 2025) differs by allowing a pre-trained model to adapt to a target domain using only unlabeled test data. Our research centers on online test-time adaptation (OTTA) (Wang et al., 2024), a scenario where test data from the target domain is presented sequentially, requiring real-time adaptation.

Recent advancements in OTTA have explored various strategies. Some methods (Wang et al., 2021; Gong et al., 2022; Mirza et al., 2022; Zhao et al., 2023) focus on batch normalization (BN) calibration since BN layers can incorporate domain-specific knowledge through the normalization statistics they learn (Li et al., 2017b). Several strategies (Zhang et al., 2022; Jing et al., 2022; Niu et al., 2023; Lee et al., 2024) place emphasis on entropy minimization, aiming to encourage more confident and distinct predictions by reducing the uncertainty in the model's output. Numerous approaches focus on pseudo-labeling, where the model generates pseudo labels for unlabeled data and performs self-training to improve its performance. Iwasawa & Matsuo (2021) create a pseudo-prototype for each class and classify new samples based on the distances. Goyal et al. (2022) introduce a self-training method that utilizes a specialized soft label known as the conjugate pseudo label. Shin et al. (2022) introduce a selective fusion strategy to ensemble predictions from multiple modalities. Meanwhile, Yang et al. (2022) generate soft pseudo labels by averaging the predictions of neighboring samples stored in a memory bank. Döbler et al. (2023) use symmetric cross-entropy loss to enforce prediction consistency between the teacher and student model. Similarly, Jang et al. (2023) aim to reduce divergence between predictions from prototype-based and neighbor-based classifiers. Wang et al. (2023) propose test-time self-distillation for feature uniformity and memorized spatial local clustering for feature alignment. Sun et al. (2024) combine prototype-based and nearest-neighbor methods through a prototype graph model to enhance pseudo-label generation.

**Uncertainty modeling.** The previous methods within unsupervised domain adaptation or source-free domain adaptation build up the uncertainty at the model or sample level. For example, at the model level, Zhan et al. (2023) leverage Monte Carlo dropout, and Lao et al. (2021) apply deep ensembles. At the sample level, Liang et al. (2020b); Roy et al. (2022) generate weights using prediction entropy, while Litrico et al. (2023) derive them by analyzing consensus among neighboring samples. Tan et al. (2024) capture both model uncertainty and sample uncertainty using a sub-network. Our proposed method models the uncertainty at the label level through candidate pseudo-label sets, where the uncertainty is directly manifested through the cardinality of the candidate pseudo-label set. Compared to existing approaches, our method imposes no constraints on pre-trained models and does not require additional computations during inference, making it particularly suitable for OTTA scenarios.

# 3 PROPOSED METHOD

## 3.1 PRELIMINARIES

To begin with, some necessary notations are briefly introduced. Let $\mathcal{X} = \mathbb{R}^q$ denote the $q$-dimensional instance space, $\mathcal{Y} = \{1, 2, \ldots, c\}$ be the label space with $c$ class labels, and $\Delta^{c-1}$ be the $c$-dimensional probability simplex. A pre-trained predictive model $f : \mathcal{X} \mapsto \Delta^{c-1}$ with parameters $\boldsymbol{\Theta}$ is initialized to $\boldsymbol{\Theta}^0$ on a dataset from the source domain $S$, i.e., $\mathcal{D}_S = \{(\boldsymbol{x}_i, y_i) | 1 \le i \le n\}$ where $\boldsymbol{x}_i \in \mathcal{X}$ denotes the $q$-dimensional instance, $y_i \in \mathcal{Y}$ denotes the correct label annotated to $\boldsymbol{x}_i$ and each example $(\boldsymbol{x}_i, y_i)$ is sampled from the distribution $p_S(\boldsymbol{x}, y)$. During test-time inference, the predictive model $f(\cdot; \boldsymbol{\Theta})$ streamingly receives $R$ mini-batch datasets from the target domain $T$. At the $r$-th step, the received dataset is denoted by $\mathcal{D}_T^r = \{\boldsymbol{x}_i^r | 1 \le i \le m^r\}$ where the received instance $\boldsymbol{x}_i^r \in \mathcal{X}$ and its unobserved correct label $y_i^r$ are from the misaligned distribution $p_T(\boldsymbol{x}, y)$ with $p_S(\boldsymbol{x}, y)$, i.e., $p_T(\boldsymbol{x}, y) \ne p_S(\boldsymbol{x}, y)$. The goal of OTTA is to continuously update the parameters $\boldsymbol{\Theta}$ of the pre-trained predictive model $f$ to obtain the maximization of the following cumulative accuracy on $R$ mini-batch datasets from the target domain $T$:

$$\text{Acc}(f, T) = \frac{\sum_{r=1}^{R} \sum_{i=1}^{m^r} \mathbb{I}[y_i^r = \arg\max_{j \in \mathcal{Y}} f_j(\boldsymbol{x}_i^r; \boldsymbol{\Theta}^r)]}{\sum_{r=1}^{R} m^r}, \tag{1}$$

where $\boldsymbol{\Theta}^r$ is the updated parameters of the model $f$ at the $r$-th step, and $\mathbb{I}[\cdot]$ is the indicator function. To achieve this goal, the process of progressive label enhancement in our framework introduces pseudo-labels through soft labels output by the predictive model to handle the absence of correct labels during adaptation. The pseudo-label of the testing instance $\boldsymbol{x}_i^r$ is denoted by a logical label vector $\boldsymbol{l}_i^r = [l_i^{r,1}, l_i^{r,2}, \ldots, l_i^{r,c}]^\top \in \{0, 1\}^c$, and the corresponding soft label is denoted by a vector $\boldsymbol{d}_i^r = [d_i^{r,1}, d_i^{r,2}, \ldots, d_i^{r,c}]^\top = [f_1(\boldsymbol{x}_i^r; \boldsymbol{\Theta}^r), f_2(\boldsymbol{x}_i^r; \boldsymbol{\Theta}^r), \ldots, f_c(\boldsymbol{x}_i^r; \boldsymbol{\Theta}^r)] \in [0, 1]^c$ satisfying $\sum_{j=1}^{c} d_i^{r,j} = 1$. When $\sum_{j=1}^{c} l_i^{r,j} = 1$, the pseudo-label of the testing instance $\boldsymbol{x}_i^r$ is a one-hot pseudo-label. When $1 < \sum_{j=1}^{c} l_i^{r,j} < c$, the pseudo-label of the testing instance $\boldsymbol{x}_i^r$ is a candidate pseudo-label set. Besides, we prepare a buffer $\mathcal{B}^r$ initialized by $\mathcal{B}^0 = \emptyset$ with maximum size $K$ to save the received instances not used at the $r$-th step.

## 3.2 OVERVIEW

Our framework trains the predictive model with the optimization objective on the split confident subset with one-hot pseudo-labels and the uncertain subset with candidate pseudo-label sets, which are constructed by a selective enhancement strategy using a dynamic threshold. This strategy follows the derived proposition about uncertainty information. Also, we select and incorporate the received instances that are not used at the current optimization step into the prepared buffer for subsequent use. Finally, as the predictive model becomes increasingly aligned with the target domain distribution, the threshold will decrease to progressively refine the pseudo-labels. In this way, we unleash the power of pseudo-labels in OTTA and improve the performance of the predictive model. Theoretically, we demonstrate that our framework achieves a tighter generalization bound by incorporating more target domain instances and evaluates the effectiveness of pseudo-labels by quantifying them through the pseudo-label error term.

### 3.3 THE PASLE FRAMEWORK

First of all, we introduce the overall objective $\mathcal{L}$ of the PASLE framework employed to optimize the predictive model parameters $\boldsymbol{\Theta}^{r-1}$ at $r$-th step:

$$\mathcal{L} = \frac{1}{|\mathcal{D}_H^r|} \sum_{\boldsymbol{x} \in \mathcal{D}_H^r} \ell(f(\boldsymbol{x}; \boldsymbol{\Theta}^{r-1}), \boldsymbol{l}) + \frac{1}{|\mathcal{D}_M^r|} \sum_{\boldsymbol{x}' \in \mathcal{D}_M^r} \ell'(f(\boldsymbol{x}'; \boldsymbol{\Theta}^{r-1}), \boldsymbol{l}'), \tag{2}$$

where $\ell(\cdot, \cdot)$ is the loss function to handle each instance $\boldsymbol{x}$ from the confident subset $\mathcal{D}_H^r$ with its one-hot pseudo-label $\boldsymbol{l}$ while $\ell'(\cdot, \cdot)$ is the loss function to handle each instance $\boldsymbol{x}'$ from the uncertain subset $\mathcal{D}_M^r$ with its candidate pseudo-label $\boldsymbol{l}'$, and $|\cdot|$ represents the cardinality of a set. Practically, $\ell(\cdot, \cdot)$ is instantiated as the cross-entropy loss, which has been widely used in supervised learning tasks, while $\ell'(\cdot, \cdot)$ is instantiated as the classifier-consistent loss proposed by Feng et al. (2020):

$$\ell'(f(\boldsymbol{x}'; \boldsymbol{\Theta}^{r-1}), \boldsymbol{l}') = -\log \sum_{j=1}^{c} l'^j f_j(\boldsymbol{x}'; \boldsymbol{\Theta}^{r-1}), \tag{3}$$

which has been validated to deal with candidate labels effectively.

Note that Eq. (2) model the uncertainty information based on the confident subset $\mathcal{D}_H^r$ and uncertain subset $\mathcal{D}_M^r$, which are split from the received dataset $\mathcal{D}_T^r$ and the buffer dataset $\mathcal{B}^{r-1}$. The confident set $\mathcal{D}_H^r$ contains the instances whose correct labels could be determined, while the uncertain set $\mathcal{D}_M^r$ contains the instances whose correct labels could be only determined among some candidate labels. Hence, we assign the one-hot pseudo-label $\boldsymbol{l}$ to each instance $\boldsymbol{x} \in \mathcal{D}_H^r$, and the candidate pseudo-label $\boldsymbol{l}'$ to each instance $\boldsymbol{x}' \in \mathcal{D}_M^r$.

Then, a selective label enhancement strategy is induced by the following proposition to construct the confident subset $\mathcal{D}_H^r$ and uncertain subset $\mathcal{D}_M^r$ by utilizing soft pseudo-labels, generating the one-hot pseudo-label $\boldsymbol{l}$ for each instance $\boldsymbol{x} \in \mathcal{D}_H^r$ and candidate pseudo-label $\boldsymbol{l}'$ for each instance $\boldsymbol{x}' \in \mathcal{D}_M^r$.

Let $\boldsymbol{\Theta}^\star$ be the optimal parameters learned from supervised data from the target domain $T$, i.e., $\boldsymbol{\Theta}^\star = \arg\min_{\boldsymbol{\Theta}} \mathbb{E}_{(\boldsymbol{x}, y) \sim p_T(\boldsymbol{x}, y)} \mathbb{I}[\arg\max_{j \in \mathcal{Y}} f_j(\boldsymbol{x}; \boldsymbol{\Theta}) \neq y)]$, $f(\boldsymbol{x}; \boldsymbol{\Theta}^\star)$ be the class-posterior probabilities for the instance $\boldsymbol{x}$ on the target domain $T$, i.e., $f_j(\boldsymbol{x}; \boldsymbol{\Theta}^\star) = p_T(y = j|\boldsymbol{x})$, $p$ represent the class the highest score of the soft label $\boldsymbol{d}_i^r$, i.e., $p = \arg\max_{j \in \mathcal{Y}} d_i^{r,j}$, and $q$ denote the class with the second-highest score of the soft label $\boldsymbol{d}_i^r$, i.e., $q = \arg\max_{j \in \mathcal{Y}, j \neq p} d_i^{r,j}$. We provide the following proposition.

**Proposition 1** *Assume that at adaptation step $r$, the difference between $f(\cdot; \boldsymbol{\Theta}^r)$ and $f(\cdot; \boldsymbol{\Theta}^\star)$ on the instance $\boldsymbol{x}_i^r$ is bounded by $\frac{1}{2}\tau(r)$, i.e., $|f_j(\boldsymbol{x}_i^r; \boldsymbol{\Theta}^r) - f_j(\boldsymbol{x}_i^r; \boldsymbol{\Theta}^\star)| \leq \frac{1}{2}\tau(r), \forall j \in \mathcal{Y}$. For any $\boldsymbol{x}_i^r$ in the unlabeled data batch $\mathcal{D}_T^r$, if $d_i^{r,p} - d_i^{r,j} > \tau(r)$, then $j$ cannot be the correct label of $\boldsymbol{x}_i^r$. Furthermore, if $d_i^{r,p} - d_i^{r,q} > \tau(r)$, then $p$ is the correct label of $\boldsymbol{x}_i^r$.*

The detailed proof can be found in the Appendix A. On the one hand, Proposition 1 provides a condition under which the label with the highest predicted score is guaranteed to be correct, allowing for the accurate assignment of a pseudo-single label to a part of the unlabeled samples. Hence, we could split a confident subset $\mathcal{D}_H^r$ from the received dataset $\mathcal{D}_T^r$ and the buffer dataset $\mathcal{B}^{r-1}$ based on the soft label of each instance and the threshold $\tau(r)$ at the $r$-th step:

$$\mathcal{D}_H^r = \{\boldsymbol{x} | \boldsymbol{x} \in \mathcal{D}_T^r \cup \mathcal{B}^{r-1}, d^p - d^q > \tau(r)\}. \tag{4}$$

After splitting the confident subset $\mathcal{D}_H^r$, we generate the one-hot pseudo-label $\boldsymbol{l}$ for each instance $\boldsymbol{x} \in \mathcal{D}_H^r$ as follows:

$$l^j = \begin{cases} 1, & \text{if } j = p \text{ and } d^p - d^q > \tau(r), \\ 0, & \text{otherwise}. \end{cases} \tag{5}$$

On the other hand, Proposition 1 provides a condition under which the label with the low predicted score is guaranteed to be incorrect, allowing for the ambiguous assignment of candidate pseudo-labels to a part of the unlabeled samples. Hence, we could split an uncertain subset $\mathcal{D}_M^r$ from the

---

**Algorithm 1** PASLE Algorithm

---

**Require:** The pre-trained predictive model $f(\cdot; \boldsymbol{\Theta}^0)$, total adaption steps $R$, initial threshold $\tau_{\text{start}}$, end threshold $\tau_{\text{end}}$, threshold descend constant $\tau_{\text{des}}$, buffer maximum size $K$;
 1: $\tau(1) \leftarrow \tau_{\text{start}}$;
 2: Initialize the buffer $\mathcal{B}^0$ as empty;
 3: **for** $r = 1, 2, ...R$ **do**
 4:     Obtain the confident subset $\mathcal{D}_H^r$ and uncertain subset $\mathcal{D}_M^r$ according to Eq. (4) and Eq. (6);
 5:     Generate the one-hot pseudo-label $\boldsymbol{l}$ for each instance $\boldsymbol{x} \in \mathcal{D}_H^r$ using Eq. (5) and the candidate pseudo-label $\boldsymbol{l}'$ for each instance $\boldsymbol{x}' \in \mathcal{D}_M^r$ using Eq. (7);
 6:     Optimize the predictive model parameters $\boldsymbol{\Theta}^{r-1}$ to $\boldsymbol{\Theta}^r$ based on Eq. (2);
 7:     Update the buffer $\mathcal{B}^{r-1}$ to $\mathcal{B}^r$ following Eq. (8);
 8:     Adjust the threshold according to Eq. (9);
 9: **end for**
**Ensure:** The predictive model $f(\cdot; \boldsymbol{\Theta})$.

---

received dataset $\mathcal{D}_T^r$ and the buffer dataset $\mathcal{B}^{r-1}$ based on the soft label of each instance and the threshold $\tau(r)$ at the $r$-th step:

$$\mathcal{D}_M^r = \{\boldsymbol{x}'|\boldsymbol{x}' \in \mathcal{D}_T^r \cup \mathcal{B}^{r-1}, \exists j \in \mathcal{Y}, d'^p - d'^j > \tau(r)\}. \tag{6}$$

After splitting the uncertain subset $\mathcal{D}_M^r$, we generate the candidate pseudo-label $\boldsymbol{l}'$ for each instance $\boldsymbol{x}' \in \mathcal{D}_M^r$ based on the corresponding soft label $\boldsymbol{d}'$ and the threshold $\tau(r)$ at the $r$-th step as follows:

$$l'^j = \begin{cases} 0, & \text{if } d'^p - d'^j > \tau(r), \\ 1, & \text{otherwise.} \end{cases} \tag{7}$$

Besides, for the received instances which are not used in Eq. (2), i.e., $(\mathcal{D}_T^r \cup \mathcal{B}^{r-1}) \setminus (\mathcal{D}_H^r \cup \mathcal{D}_M^r)$, we intend to store them in the new buffer $\mathcal{B}^r$. If the number of samples to be stored exceeds the buffer's maximum size during the algorithm's operation, we prioritize retaining the samples with the top-$K$ largest margins, as these are likely to contribute to the model's updates earlier. The margin of an instance $\boldsymbol{x}$ is defined as $\rho(\boldsymbol{x}) = \boldsymbol{d}^p - \boldsymbol{d}^q$. Denote $\rho_K$ the top-$K$ largest margin of the instance $\boldsymbol{x} \in (\mathcal{D}_T^r \cup \mathcal{B}^{r-1}) \setminus (\mathcal{D}_H^r \cup \mathcal{D}_M^r)$ at adaptation step $r$. Then, the sample selection process can be formulated as follows:

$$\mathcal{B}^r = \{\boldsymbol{x}|\boldsymbol{x} \in (\mathcal{D}_T^r \cup \mathcal{B}^{r-1}) \setminus (\mathcal{D}_H^r \cup \mathcal{D}_M^r), \rho(\boldsymbol{x}) > \rho_K\}, \tag{8}$$

where the buffer's maximum capacity $K$ is restricted to a quarter of the target domain batch size in practice, considering resource consumption.

Finally, since the model becomes increasingly aligned with the target domain distribution as adaptation progresses, the gap between the scoring function $f(\cdot; \boldsymbol{\Theta}^r)$ and the class-posterior probability $f(\cdot; \boldsymbol{\Theta}^\star)$ is supposed to gradually decrease. Hence, to improve the reliability of pseudo-labels, the threshold $\tau(r)$ is manually reduced over time. In practice, $\tau(r)$ is initialized at a starting value $\tau_{\text{start}}$ when $r = 1$ and is gradually decreased by a constant value $\tau_{\text{des}}$ each step. This process continues until $\tau(r)$ reaches a predefined lower bound $\tau_{\text{end}}$. The adjustment of $\tau(r)$ can be formulated as:

$$\tau(r) = \max\{\tau(r-1) - \tau_{\text{des}}, \tau_{\text{end}}\}. \tag{9}$$

In this way, the PASLE framework successfully leverages the uncertainty information and flexibly tunes pseudo-labels according to the model's evolving adaptation capability, thereby unleashing the potential of pseudo-labels and improving the performance of TTA. The algorithmic description of PASLE is presented in Algorithm 1.

### 3.4 THEORETICAL ANALYSIS

#### 3.4.1 A GENERALIZATION BOUND FOR TTA

We theoretically establish a generalization bound for TTA that by incorporating a greater number of target domain samples with effective supervision, a tighter generalization bound can be achieved.

Suppose that at the $r$-th step of adaptation, the classifier $h$ receives $m^r$ samples. Given the possibility of overlap between the source domain and the target domain, we assume $(1 - \beta)m^r$ samples are drawn from $S$, and the remaining $\beta m^r$ samples are drawn from $T$. The goal of the classifier is to find a hypothesis that minimizes the target error $\epsilon_T(h)$. We focus on classifiers that minimize a convex combination of the empirical errors from the source and target domains (Ben-David et al., 2010), defined as:

$$\hat{\epsilon}_\alpha(h) = \alpha \hat{\epsilon}_T(h) + (1 - \alpha)\hat{\epsilon}_S(h), \tag{10}$$

where $\alpha \in [0, 1]$. The corresponding weighted combination of the true source and target errors is denoted by $\epsilon_\alpha(h)$. To quantify the distributional difference between the source domain and target domain, we use the disparity discrepancy introduced by Zhang et al. (2019).

**Definition 1** *Given a hypothesis space $\mathcal{H}$ and a specific classifier $h \in \mathcal{H}$, the Disparity Discrepancy induced by $h' \in \mathcal{H}$ is defined by*

$$
\begin{aligned}
d_{h,\mathcal{H}}(S, T) &\triangleq \sup_{h' \in \mathcal{H}} \left( \mathrm{disp}_T\left(h', h\right) - \mathrm{disp}_S\left(h', h\right) \right) \\
&= \sup_{h' \in \mathcal{H}} \left( \mathbb{E}_T \mathbb{I}\left[h' \neq h\right] - \mathbb{E}_S \mathbb{I}\left[h' \neq h\right] \right).
\end{aligned} \tag{11}
$$

Based on the above assumptions and definition, we derive the following theorem.

**Theorem 1** *Let $\mathcal{H}$ be a hypothesis space of VC dimension $d$. Let $\widehat{S}$ and $\widehat{T}$ be unlabeled sample sets of size $m'$ each, drawn from $S$ and $T$ respectively. A batch of samples of size $m^r$ is generated by random sampling at the $r$-th step of adaptation. Given the possibility of overlap between the source domain and the target domain, we assume $(1 - \beta)m^r$ samples are drawn from $S$, and the remaining $\beta m^r$ samples are drawn from $T$, which are then labeled with the true labeling function. For simplicity in theoretical analysis, we allocate the loss weight proportionally to the number of samples from each domain, specifically setting $\alpha$ and $\beta$ to be equal. If $\hat{h} \in \mathcal{H}$ is the empirical minimizer of $\hat{\epsilon}_\alpha(h)$ on this batch and $h_T^* = \min_{h \in \mathcal{H}} \epsilon_T(h)$ is the target error minimizer, then for any $\delta > 0$, with probability at least $1 - 2\delta$,*

$$
\begin{aligned}
\epsilon_T(\hat{h}) \leq{}& \epsilon_T\left(h_T^*\right) + 4\sqrt{\frac{2d \log(2(m^r + 1)) + 2 \log\left(\frac{4}{\delta}\right)}{m^r}} \\
&+ 2(1 - \beta)\left(d_{h,\mathcal{H}}(\widehat{S}, \widehat{T}) + 2\sqrt{\frac{2d \log \frac{em'}{4d}}{m'}} + 2\sqrt{\frac{\log \frac{2}{\delta}}{2m'}} + \lambda\right),
\end{aligned} \tag{12}
$$

*where $\lambda = \mathrm{err}_S\left(h^*\right) + \mathrm{err}_T\left(h^*\right)$ and $h^* \triangleq \arg\min_{h \in \mathcal{H}} \left\{\mathrm{err}_S(h) + \mathrm{err}_T(h)\right\}$.*

The proof of Theorem 1 is provided in Appendix B. Theorem 1 provides a generalization bound on the target domain for the empirical minimizer on the given batch. Suppose we also have some target domain samples annotated with effective supervisory information provided alongside the well-annotated current data batch to guide the model's adaptation. In that case, the empirical minimizer's generalization bound on the target domain becomes tighter. Since the growth rate of the upper bound's second term is $\mathcal{O}(\sqrt{\frac{\log m}{m}})$, it decreases as more samples are incorporated into training. Moreover, as the number of target domain samples increases, $\beta$ will also increase, resulting in a reduction of the third term in the upper bound.

### 3.4.2 PSEUDO-LABEL EFFECTIVENESS QUANTIFICATION

Next, we assess the effectiveness of pseudo-labels by quantifying them through pseudo-label error terms for TTA. Assume that during test-time inference, the predictive model streamingly receives $R$ mini-batch data from the target domain $T$, accumulating a dataset $\mathcal{D}_T^R$ over $R$ mini-batches, with a total sample size of $N^R$. For a target domain sample $\boldsymbol{x}$, let its Bayes class-probability distribution be denoted as $\boldsymbol{p} = [P\left(y_1 \mid \boldsymbol{x}\right), P\left(y_2 \mid \boldsymbol{x}\right), \ldots, P\left(y_c \mid \boldsymbol{x}\right)]$, and its supervision information provided by the algorithm be denoted as $\boldsymbol{q}$ (here it refers to the label distribution). We have the following theorem.

**Theorem 2** *Suppose the loss function $\ell$ is bounded by $M$, i.e., $M = \sup_{\boldsymbol{x} \in \mathcal{X}, f \in \mathcal{F}, y_j \in \mathcal{Y}} \ell(f(\boldsymbol{x}), y)$. Fix a hypothesis class $\mathcal{F}$ of predictors $f : \mathcal{X} \mapsto \mathbb{R}^c$, with induced class $\mathcal{H} \subset [0,1]^{\mathcal{X}}$ of functions $h(\boldsymbol{x}) = \ell(f(\boldsymbol{x}_i), \boldsymbol{q})$. Suppose $\mathcal{H}$ has uniform covering number $\mathcal{N}_{inf}$. Then for any $\delta \in (0,1)$, with probability at least $1 - \delta$,*

$$R(f) - \widehat{R}(f) \leq M\sqrt{c} \cdot (\mathbb{E}[\|\boldsymbol{q} - \boldsymbol{p}\|_2]) + \mathcal{O}\left(\sqrt{\mathbb{V}(f) \cdot \frac{\log \frac{\mathcal{M}_{N^R}}{\delta}}{N^R}} + \frac{\log \frac{\mathcal{M}_{N^R}}{\delta}}{N^R}\right), \qquad (13)$$

*where $\mathcal{M}_{N^R} = \mathcal{N}_{\inf}\left(\frac{1}{N^R}, \mathcal{H}, 2N^R\right)$, and $\mathbb{V}(f)$ is the empirical variance of the loss values.*

The proof of Theorem 2 is provided in Appendix C. Theorem 2 demonstrates that as the target domain samples' label distribution $\boldsymbol{q}$ provided by the algorithm becomes closer to the Bayes class-probability distribution $\boldsymbol{p}$, the gap between the empirical risk and the expected risk on the accumulated dataset $\mathcal{D}_T^R$ will decrease. The effectiveness of the supervision information can be quantified by the degree of closeness between its corresponding label distribution and the Bayes class-probability distribution and the pseudo-label error term is $\mathbb{E}[\|\boldsymbol{q} - \boldsymbol{p}\|_2]$. Our algorithm provides one-hot pseudo-labels when it is more certain about the samples and a candidate pseudo-label set when it is uncertain. These actions can make the corresponding pseudo-labels' label distribution closer to the Bayes class-probability distribution, thereby making the empirical risk more closely aligned with the expected risk and thus better guiding the model toward adaptation to the target domain.

## 4 EXPERIMENTS

### 4.1 DATASETS

Following recent works in OTTA (Jang et al., 2023; Wang et al., 2023; Sun et al., 2024), we use domain generalization datasets and image corruption datasets to evaluate our method. We employ four domain generalization datasets including `PACS` (Li et al., 2017a), `VLCS` (Torralba & Efros, 2011), `OfficeHome` (Venkateswara et al., 2017), and `DomainNet` (Peng et al., 2019). `DomainNet` is a large-scale dataset with 586,575 images across 345 classes, covering six domains: clipart, infograph, painting, quickdraw, real, and sketch. For source training, we designate one domain as the target and use the remaining domains as source domains. We allocated 20% of the data from the source domains for validation purposes.

Additionally, we employ two image corruption datasets: `CIFAR-10-C` and `CIFAR-100-C` (Hendrycks & Dietterich, 2019). Both datasets introduce 15 types of common image corruptions, categorized into four types: noise, blur, weather, and digital, to the test sets of `CIFAR-10` and `CIFAR-100` (Krizhevsky, 2009). We use the training sets of `CIFAR-10` and `CIFAR-100` as source domains and the highest level of corruption in `CIFAR-10-C` and `CIFAR-100-C` as target domains. The validation set partition follows the same approach as that used for the domain generalization datasets.

### 4.2 BASELINES

We compare the performance of PASLE with ten OTTA approaches.

- ERM (Vapnik, 1998): A standard approach that directly outputs the model's predictions without adaptation.
- BN (Schneider et al., 2020): A BN calibration approach that replaces the activation statistics estimated by BN layers on the training set with the statistics of the target domain images.
- TENT (Wang et al., 2021): An entropy minimization approach that optimizes channel-wise affine transformations by reducing the entropy of model predictions on target domain data.
- PL (Lee, 2013): A pseudo-labeling approach that fine-tunes a pre-trained classifier by leveraging confident pseudo-labels derived from the model's predictions.
- SHOT-IM (Liang et al., 2020a): An information maximization approach that refines the source encoding module by maximizing the mutual information between intermediate feature representations and the classifier's outputs.

Table 1: Classification accuracy of comparing methods on domain generalization datasets.

| Methods | BackBone | PACS | VLCS | OfficeHome | DomainNet | Avg. |
|---|---|---|---|---|---|---|
| ERM | | 79.37 | 75.14 | 62.43 | 35.76 | 63.18 |
| BN | | 83.08 | 68.79 | 62.29 | 34.95 | 62.28 |
| TENT | | 83.23 | 69.28 | 62.51 | 35.37 | 62.60 |
| PL | | 85.66 | 74.68 | 62.71 | 35.24 | 64.57 |
| SHOT-IM | | 83.02 | 70.80 | 63.91 | 35.92 | 63.41 |
| T3A | | 81.70 | 75.83 | 63.90 | 36.31 | 64.44 |
| TAST | ResNet-18 | 84.31 | 71.69 | 63.96 | 35.71 | 63.92 |
| TAST-BN | | 86.35 | 75.17 | 62.43 | 35.82 | 64.94 |
| TSD | | 87.88 | 75.47 | 63.42 | 35.86 | 65.66 |
| PROGRAM | | 83.57 | 71.64 | 63.35 | 35.97 | 63.63 |
| DEYO | | 86.26 | 74.91 | 63.30 | 35.37 | 64.96 |
| PASLE | | **88.16** | **77.91** | **63.99** | **36.89** | **66.74** |
| ERM | | 85.84 | 76.06 | 67.84 | 43.16 | 68.23 |
| BN | | 86.00 | 67.76 | 66.82 | 41.50 | 65.52 |
| TENT | | 86.51 | 68.41 | 67.27 | 42.38 | 66.14 |
| PL | | 85.66 | 73.80 | 67.31 | 42.38 | 67.29 |
| SHOT-IM | | 85.27 | 68.49 | 67.89 | 43.41 | 66.27 |
| T3A | | 86.54 | 76.59 | 68.85 | 44.00 | 69.00 |
| TAST | ResNet-50 | 86.94 | 67.32 | 68.70 | 42.84 | 66.45 |
| TAST-BN | | 89.47 | 75.59 | 67.97 | 43.03 | 69.02 |
| TSD | | 91.13 | 74.77 | 68.97 | 42.44 | 69.33 |
| PROGRAM | | 86.16 | 68.85 | 68.03 | 43.34 | 66.60 |
| DEYO | | 88.23 | 71.59 | 68.08 | 42.47 | 67.59 |
| PASLE | | **91.36** | **78.70** | **69.37** | **44.91** | **71.09** |

Table 2: Classification accuracy of comparing methods on image corruption datasets.

| Methods | CIFAR-10-C | CIFAR-100-C |
|---|---|---|
| ERM | 20.66 | 5.84 |
| BN | 75.33 | 43.88 |
| TENT | 75.41 | 43.93 |
| PL | 75.70 | 44.24 |
| SHOT-IM | 75.85 | 44.36 |
| T3A | 23.52 | 6.74 |
| TAST | 74.13 | 39.21 |
| TAST-BN | 74.56 | 31.84 |
| TSD | 75.14 | 44.19 |
| PROGRAM | 75.00 | 44.06 |
| DEYO | 75.74 | 44.28 |
| PASLE | **76.67** | **45.32** |

- T3A (Iwasawa & Matsuo, 2021): A pseudo-labeling approach that predicts test data labels by measuring the distances between the test samples and pseudo-prototypes.

- TAST (Jang et al., 2023): A pseudo-labeling approach that updates the model by matching the nearest neighbor-based pseudo label and a prototype-based class distribution for the test data.

- TAST-BN (Jang et al., 2023): A pseudo-labeling approach, which is a modified version of TAST and refines the model by adjusting the parameters of the BN layers.

Table 3: Classification accuracy (mean ± std) of PASLE and its variant on target domains of the `OfficeHome` dataset. The best performance is shown in boldface.

| Domain | PASLE | PASLE-NC |
|--------|-------|----------|
| A | **57.25±0.75** | 56.11±0.92 |
| C | **51.30±0.41** | 50.85±0.68 |
| P | **73.31±1.04** | 72.13±1.43 |
| R | **74.10±0.20** | 73.87±0.45 |

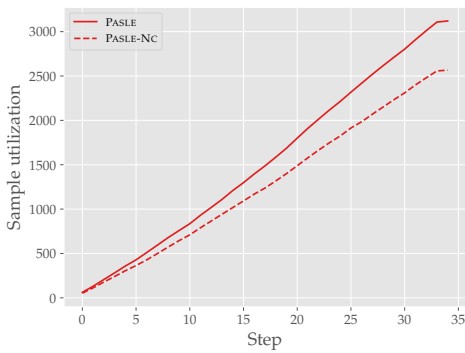

Figure 1: Sample utilization during testing.

- TSD (Wang et al., 2023): A pseudo-labeling approach that employs a dynamic memory bank to compute class prototypes and generate pseudo-labels for refining the model.

- PROGRAM (Sun et al., 2024): A pseudo-labeling approach that leverages a graph structure that connects prototypes with test samples, allowing information to flow between them and enhancing the generation of pseudo-labels.

- DEYO (Lee et al., 2024): An entropy minimization approach that updates the model by minimizing the model output's entropy of samples with low entropy and high PLPD value concurrently.

We evaluate all methods on domain generalization benchmarks using ResNet-18 and ResNet-50 models (He et al., 2016), both equipped with batch normalization (Ioffe & Szegedy, 2015). For image corruption benchmarks, we employ ResNet-18 as the backbone model.

For source training, the models are trained using the Adam optimizer with a learning rate of $5e^{-5}$ for domain generalization benchmarks and $1e^{-3}$ for image corruption benchmarks. All weights are initialized from ImageNet-1K (Russakovsky et al., 2015) pre-trained models. We select the final pre-trained model with the highest validation accuracy.

During testing, we also utilize the Adam optimizer to update all trainable layers without the need for a specific selection. The batch size for the online target domain data is set to 128, with the buffer capacity $K$ set to one-fourth of the batch size, i.e., 32. The learning rate is selected from the range between $1e^{-3}$ and $1e^{-6}$. The value of $\tau_{\text{start}}$ is determined by the number of classes in each dataset: for example, `VLCS` contains 5 classes, while `DomainNet` has 345 classes, leading to different $\tau_{\text{start}}$ values for each dataset. The threshold gap, represented as $|\tau_{\text{start}} - \tau_{\text{end}}|$, is consistently set at 0.1. Furthermore, $\tau_{\text{des}}$ is uniformly set to $1e^{-3}$ for all datasets, except for the large-scale dataset `DomainNet`, where it is adjusted to $1e^{-4}$. It is essential to highlight that all hyperparameters for the OTTA setting are finalized prior to accessing any test samples. We meticulously select the most suitable hyperparameters for each algorithm based on their performance on the training domain validation datasets (Gulrajani & Lopez-Paz, 2021; Wang et al., 2023).

### 4.3 EXPERIMENTAL RESULTS

We conduct 3 trials with different random seeds, reporting both the mean and standard deviation of the metrics, with full results detailed in Appendix E. Each method's classification accuracy across datasets' target domains is summarized in Table 1 and Table 2. The best performance is shown in boldface, and the second-best result is underlined. It is impressive to observe that:

- PASLE achieves the best performance across all benchmark datasets and network architectures, outperforming all the compared approaches.

- PASLE consistently enhances the performance of the classifier on domain generalization benchmarks, achieving an average improvement of 5.63% for ResNet-18 and 4.19% for ResNet-50.

- PASLE outperforms the second-best methods on image corruption benchmarks, with an average performance gain of 1.08% on `CIFAR-10-C` and 2.16% on `CIFAR-100-C`.

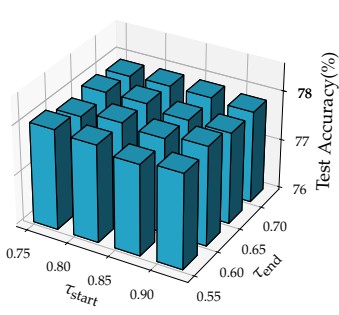 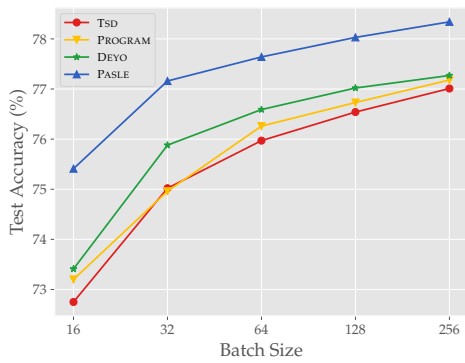

(a) The sensitivity of $\tau_{\text{start}}$ and $\tau_{\text{end}}$ on CIFAR-10-C.  (b) The sensitivity of batch size on CIFAR-10-C.

Figure 2: The parameter sensitivity analysis for PASLE.

- Especially, we find that PASLE significantly improves the performance of the pre-trained classifier on the DomainNet benchmark, which is a large-scale dataset, indicating that PASLE holds strong potential for real-world applications.

## 4.4 FURTHER ANALYSIS

To demonstrate the effectiveness of the candidate pseudo-labels in PASLE, we conduct an ablation study using a vanilla variant, PASLE-NC. In PASLE-NC, all samples annotated with candidate pseudo-labels are excluded from model updates. As shown in Table 3, PASLE outperforms PASLE-NC on all target domains of the OfficeHome dataset with ResNet-18, highlighting the importance of candidate pseudo-labels in further improving performance during the testing phase.

Furthermore, Figure 1 shows the sample utilization during the testing process for both PASLE and PASLE-NC on the clipart domain of OfficeHome dataset. It is evident that PASLE utilizes more effectively labeled samples than PASLE-NC as testing progresses, which further explains PASLE's better test-time adaptation performance.

Besides, we perform a parameter sensitivity analysis to examine the impact of two hyperparameters $\tau_{\text{start}}$ and $\tau_{\text{end}}$ on our algorithm using the shot noise corruption of CIFAR-10-C dataset. $\tau_{\text{start}}$ and $\tau_{\text{end}}$ are assigned various values as illustrated in Figure 2(a), with $\tau_{\text{des}}$ being set to $\frac{\tau_{\text{start}} - \tau_{\text{end}}}{R}$ specifically for the purpose of sensitivity analysis. Clearly, the performance of PASLE remains relatively stable across a broad range of each hyperparameter. This robustness is highly desirable, as the PASLE framework consistently delivers reliable test-time adaptation performance.

Figure 2(b) presents the average accuracy of various methods across different batch sizes on shot noise corruption of CIFAR-10-C dataset. As depicted in the figure, our approach consistently outperforms the other methods under varying batch sizes. This robust performance makes PASLE well-suited for deployment in practical scenarios, where the batch size of real-world data streams can fluctuate significantly.

More discussions of PASLE can be found in the appendix D.

## 5 CONCLUSION

In this paper, we focus on test-time adaptation and introduce a novel framework, PASLE, which selectively assigns one-hot pseudo-labels to confident test samples and candidate pseudo-label sets to uncertain test samples through a label enhancement process. The model is then progressively trained on both certain and uncertain pseudo-labeled target data. Throughout this process, the candidate pseudo-label sets for uncertain samples are dynamically refined by leveraging the model's evolving adaptation to the target domain, which is continuously monitored during training. Experiments on various benchmark datasets demonstrate the effectiveness and robustness of the proposed approach.

## ACKNOWLEDGMENTS

This research was supported by the Jiangsu Science Foundation (BG2024036), the National Science Foundation of China (62206050, 62125602, and U24A20324), the Fundamental Research Funds for the Central Universities (2242024k30035), and the Big Data Computing Center of Southeast University.

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

## A  PROOFS OF PROPOSITION 1

According to the assumption of Proposition 1, we have the following bounds for $f_p(\boldsymbol{x}_i^r; \boldsymbol{\Theta}^r)$ and $f_j(\boldsymbol{x}_i^r; \boldsymbol{\Theta}^r)$:

$$f_p(\boldsymbol{x}_i^r; \boldsymbol{\Theta}^\star) - \frac{1}{2}\tau(r) \leq f_p(\boldsymbol{x}_i^r; \boldsymbol{\Theta}^r) \leq f_p(\boldsymbol{x}_i^r; \boldsymbol{\Theta}^\star) + \frac{1}{2}\tau(r), \tag{14}$$

$$f_j(\boldsymbol{x}_i^r; \boldsymbol{\Theta}^\star) - \frac{1}{2}\tau(r) \leq f_j(\boldsymbol{x}_i^r; \boldsymbol{\Theta}^r) \leq f_p(\boldsymbol{x}_i^r; \boldsymbol{\Theta}^\star) + \frac{1}{2}\tau(r). \tag{15}$$

Subtract inequality 15 from inequality 14, we can deduce that:

$$d_i^{r,p} - d_i^{r,j} = f_p(\boldsymbol{x}_i^r; \boldsymbol{\Theta}^r) - f_j(\boldsymbol{x}_i^r; \boldsymbol{\Theta}^r) \leq f_p(\boldsymbol{x}_i^r; \boldsymbol{\Theta}^\star) - f_j(\boldsymbol{x}_i^r; \boldsymbol{\Theta}^\star) + \tau(r). \tag{16}$$

Now, if $d_i^{r,p} - d_i^{r,j} > \tau(r)$, then it follows that:

$$\tau(r) < f_p(\boldsymbol{x}_i^r; \boldsymbol{\Theta}^\star) - f_j(\boldsymbol{x}_i^r; \boldsymbol{\Theta}^\star) + \tau(r), \tag{17}$$

which simplifies to:

$$f_j(\boldsymbol{x}_i^r; \boldsymbol{\Theta}^\star) < f_p(\boldsymbol{x}_i^r; \boldsymbol{\Theta}^\star) \tag{18}$$

Therefore, $j$ cannot be the true label of $\boldsymbol{x}_i^r$.

If $d_i^{r,p} - d_i^{r,q} > \tau(r)$, then $q$ cannot be the true label of $\boldsymbol{x}_i^r$. Moreover, for any $j \in \mathcal{Y} \setminus \{p, q\}$, it follows that $d_i^{r,p} - d_i^{r,j} > \tau(r)$, since $d_i^{r,j} < d_i^{r,q}$. As a result, none of the labels in $\mathcal{Y} \setminus \{p\}$ can be the true label of $\boldsymbol{x}_i^r$. Therefore, given $d_i^{r,p} - d_i^{r,q} > \tau(r)$, $p$ must be the true label of $\boldsymbol{x}_i^r$. The proof is completed.

## B  PROOFS OF THEOREM 1

Suppose that at the $r$-th step of adaptation, the classifier $h$ receives $m^r$ samples. Given that the source domain and target domain may overlap, we assume $(1 - \beta)m^r$ samples are drawn from $S$, and the remaining $\beta m^r$ samples are drawn from $T$. The goal of the classifier is to find a hypothesis that minimizes the target error $\epsilon_T(h)$. We focus on classifiers that minimize a convex combination of the empirical errors from the source and target domains (Ben-David et al., 2010), defined as:

$$\hat{\epsilon}_\alpha(h) = \alpha\hat{\epsilon}_T(h) + (1 - \alpha)\hat{\epsilon}_S(h), \tag{19}$$

where $\alpha \in [0, 1]$. The corresponding weighted combination of the true source and target errors is denoted by $\epsilon_\alpha(h)$.

To quantify the distributional difference between the two domains, we use the disparity discrepancy introduced by Zhang et al. (2019).

**Definition 1** *Given a hypothesis space $\mathcal{H}$ and a specific classifier $h \in \mathcal{H}$, the Disparity Discrepancy induced by $h' \in \mathcal{H}$ is defined by*

$$\begin{aligned} d_{h,\mathcal{H}}(S, T) &\triangleq \sup_{h' \in \mathcal{H}} \left( \mathrm{disp}_T(h', h) - \mathrm{disp}_S(h', h) \right) \\ &= \sup_{h' \in \mathcal{H}} \left( \mathbb{E}_T \mathbb{I}[h' \neq h] - \mathbb{E}_S \mathbb{I}[h' \neq h] \right). \end{aligned} \tag{20}$$

In domain adaptation theory (Ben-David et al., 2010), the symmetric difference hypothesis space is commonly defined as follows:

**Definition 2** *Define $\mathcal{H}\Delta\mathcal{H} \triangleq \{h \mid h = h_1 \otimes h_2, h_1, h_2 \in \mathcal{H}\}$ as the symmetric difference hypothesis space of $\mathcal{H}$, where $\otimes$ stands for the XOR operator.*

We begin by establishing an upper bound on the difference between the target error $\epsilon_T(h)$ and the weighted error $\epsilon_\alpha(h)$.

**Lemma 1** *Suppose $\epsilon_S(h) \leq \epsilon_T(h)$. For any classifier h, we have:*

$$|\epsilon_\alpha(h) - \epsilon_T(h)| \leq (1-\alpha)\left[d_{h,\mathcal{H}}(S,T) + \lambda\right], \tag{21}$$

*where $\lambda = \mathrm{err}_S(h^*) + \mathrm{err}_T(h^*)$ and $h^* \triangleq \underset{h \in \mathcal{H}}{\arg\min}\{\mathrm{err}_S(h) + \mathrm{err}_T(h)\}$.*

*Proof.*

$$
\begin{aligned}
|\epsilon_\alpha(h) - \epsilon_T(h)| &= (1-\alpha)\,|\epsilon_S(h) - \epsilon_T(h)| \\
&\leq (1-\alpha)\left[\left(\mathbb{E}_T\mathbb{I}\left[h^* \neq h\right] - \mathbb{E}_S\mathbb{I}\left[h^* \neq h\right]\right) + \left(\epsilon_S(h^*) + \epsilon_T(h^*)\right)\right] \\
&\leq (1-\alpha)\left[\sup_{h' \in \mathcal{H}}\left(\mathrm{disp}_T(h',h) - \mathrm{disp}_S(h',h)\right) + \lambda\right] \\
&= (1-\alpha)\left[d_{h,\mathcal{H}}(S,T) + \lambda\right]
\end{aligned}
\tag{22}
$$

Then, we introduce an upper bound on the difference between the empirical and expected disparity discrepancy.

**Lemma 2** *Denote $\widehat{S}$ and $\widehat{T}$ the empirical distributions of datasets with $m$ and $n$ instances sampled from $S$ and $T$, respectively. For any $\delta > 0$ and binary classifier $h \in \mathcal{H}$, with probability $1 - 2\delta$,*

$$\sup_{h \in \mathcal{H}}\left(d_{h,\mathcal{H}}(\mathcal{S},\mathcal{T}) - d_{h,\mathcal{H}}(\widehat{\mathcal{S}},\widehat{\mathcal{T}})\right) \leq 2\mathfrak{R}_{m,\mathcal{S}}(\mathcal{H}\Delta\mathcal{H}) + \sqrt{\frac{\log\frac{2}{\delta}}{2m}} + 2\mathfrak{R}_{n,\mathcal{T}}(\mathcal{H}\Delta\mathcal{H}) + \sqrt{\frac{\log\frac{2}{\delta}}{2n}}. \tag{23}$$

*Proof.*

$$
\begin{aligned}
&d_{h,\mathcal{H}}(S,T) - d_{h,\mathcal{H}}(\widehat{S},\widehat{T}) \\
&= \sup_{h' \in \mathcal{H}}\left(\mathrm{disp}_T(h',h) - \mathrm{disp}_S(h',h)\right) - \sup_{h'' \in \mathcal{H}}\left(\mathrm{disp}_{\widehat{T}}(h'',h) - \mathrm{disp}_{\widehat{S}}(h'',h)\right) \\
&\leq \sup_{h' \in \mathcal{H}}\left(\mathrm{disp}_T(h',h) - \mathrm{disp}_S(h',h) - \mathrm{disp}_{\widehat{T}}(h',h) + \mathrm{disp}_{\widehat{S}}(h',h)\right) \\
&\leq \sup_{h' \in \mathcal{H}}\left(\mathrm{disp}_T(h',h) - \mathrm{disp}_{\widehat{T}}(h',h)\right) + \sup_{h'' \in \mathcal{H}}\left(\mathrm{disp}_{\widehat{S}}(h'',h) - \mathrm{disp}_S(h'',h)\right)
\end{aligned}
\tag{24}
$$

Take supremum over $h \in \mathcal{H}$, we have:

$$
\begin{aligned}
&\sup_{h \in \mathcal{H}}\left(d_{h,\mathcal{H}}(S,T) - d_{h,\mathcal{H}}(\widehat{S},\widehat{T})\right) \\
&\leq \sup_{h,h' \in \mathcal{H}}\left|\mathrm{disp}_T(h',h) - \mathrm{disp}_{\widehat{T}}(h',h)\right| + \sup_{h,h'' \in \mathcal{H}}\left|\mathrm{disp}_{\widehat{S}}(h'',h) - \mathrm{disp}_S(h'',h)\right| \\
&= \sup_{h,h' \in \mathcal{H}}\left|\mathbb{E}_T\mathbb{I}\left[h' \neq h\right] - \mathbb{E}_{\widehat{T}}\mathbb{I}\left[h' \neq h\right]\right| + \sup_{h,h' \in \mathcal{H}}\left|\mathbb{E}_S\mathbb{I}\left[h' \neq h\right] - \mathbb{E}_{\widehat{S}}\mathbb{I}\left[h' \neq h\right]\right| \\
&= \sup_{g \in \mathcal{H}\Delta\mathcal{H}}\left|\mathbb{E}_T\mathbb{I}[g \neq 1] - \mathbb{E}_{\widehat{T}}\mathbb{I}[g \neq 1]\right| + \sup_{g' \in \mathcal{H}\Delta\mathcal{H}}\left|\mathbb{E}_S\mathbb{I}[g' \neq 1] - \mathbb{E}_{\widehat{S}}\mathbb{I}[g' \neq 1]\right| \\
&= \sup_{g \in \mathcal{H}\Delta\mathcal{H}}\left|\mathbb{E}_T g - \mathbb{E}_{\widehat{T}} g\right| + \sup_{g' \in \mathcal{H}\Delta\mathcal{H}}\left|\mathbb{E}_S g' - \mathbb{E}_{\widehat{S}} g'\right| \\
&\leq 2\mathfrak{R}_{m,S}(\mathcal{H}\Delta\mathcal{H}) + \sqrt{\frac{\log\frac{2}{\delta}}{2m}} + 2\mathfrak{R}_{n,T}(\mathcal{H}\Delta\mathcal{H}) + \sqrt{\frac{\log\frac{2}{\delta}}{2n}}.
\end{aligned}
\tag{25}
$$

The following lemma provides a probabilistic bound on the difference between the empirical and true error rates of the classifier $h$.

**Lemma 3** *A batch of samples of size $m$ is generated by taking $\beta m$ samples from $T$ and $(1-\beta)m$ samples from $S$, and then labeling them with the true labeling function $g(\boldsymbol{x})$. Then we can have:*

$$\Pr\left[|\hat{\epsilon}_\alpha(h) - \epsilon_\alpha(h)| \geq \epsilon\right] \leq 2\exp\left(\frac{-2m\epsilon^2}{\frac{\alpha^2}{\beta} + \frac{(1-\alpha)^2}{1-\beta}}\right). \tag{26}$$

*Proof.* Let $x_1, \ldots, x_{\beta m}$ represent random variables corresponding to the $\beta m$ samples $\boldsymbol{x} \in T$, taking values:

$$\frac{\alpha}{\beta} |h(\boldsymbol{x}) - g(\boldsymbol{x})|. \tag{27}$$

Similarly, for the remaining $(1 - \beta)m$ samples $\boldsymbol{x} \in S$, let $x_{\beta m+1}, \ldots, x_m$ be random variables that take on the values:

$$\frac{1 - \alpha}{1 - \beta} |h(\boldsymbol{x}) - g(\boldsymbol{x})|. \tag{28}$$

The empirical weighted error $\hat{\epsilon}_\alpha(h)$ can be expressed as:

$$\hat{\epsilon}_\alpha(h) = \alpha \hat{\epsilon}_T(h) + (1 - \alpha)\hat{\epsilon}_S(h)$$
$$= \alpha \frac{1}{\beta m} \sum_{\boldsymbol{x} \in T} |h(\boldsymbol{x}) - g(\boldsymbol{x})| + (1 - \alpha)\frac{1}{(1 - \beta)m} \sum_{\boldsymbol{x} \in S} |h(\boldsymbol{x}) - g(\boldsymbol{x})| = \frac{1}{m} \sum_{i=1}^m x_i. \tag{29}$$

Applying the linearity of expectation, we have:

$$\mathbb{E}\left[\hat{\epsilon}_\alpha(h)\right] = \frac{1}{m}\left(\beta m \cdot \frac{\alpha}{\beta}\epsilon_T(h) + (1 - \beta)m \cdot \frac{1 - \alpha}{1 - \beta}\epsilon_S(h)\right)$$
$$= \alpha\epsilon_T(h) + (1 - \alpha)\epsilon_S(h) = \epsilon_\alpha(h). \tag{30}$$

It is important to note that $x_1, \ldots, x_{\beta m} \in [0, \frac{\alpha}{\beta}]$ and $x_{\beta m+1}, \ldots, x_m \in [0, \frac{1-\alpha}{1-\beta}]$. Thus, Hoeffding's Inequality can be applied as follows:

$$\Pr\left[|\hat{\epsilon}_\alpha(h) - \epsilon_\alpha(h)| \geq \epsilon\right] \leq 2\exp\left(\frac{-2m^2\epsilon^2}{\sum_{i=1}^m \text{range}^2(x_i)}\right)$$
$$= 2\exp\left(\frac{-2m^2\epsilon^2}{\beta m\left(\frac{\alpha}{\beta}\right)^2 + (1 - \beta)m\left(\frac{1-\alpha}{1-\beta}\right)^2}\right) = 2\exp\left(\frac{-2m\epsilon^2}{\frac{\alpha^2}{\beta} + \frac{(1-\alpha)^2}{1-\beta}}\right). \tag{31}$$

Building on the above definition and lemma, we derive the following theorem:

**Theorem 1** *Let $\mathcal{H}$ be a hypothesis space of VC dimension $d$. Let $\widehat{S}$ and $\widehat{T}$ be unlabeled sample sets of size $m'$ each, drawn from $S$ and $T$ respectively. A batch of samples of size $m^r$ is generated by random sampling at the $r$-th step of adaptation. Given the possibility of overlap between the source domain and the target domain, we assume $(1 - \beta)m^r$ samples are drawn from $S$, and the remaining $\beta m^r$ samples are drawn from $T$, which are then labeled with the true labeling function. For simplicity in theoretical analysis, we allocate the loss weight proportionally to the number of samples from each domain, specifically setting $\alpha$ and $\beta$ to be equal. If $\hat{h} \in \mathcal{H}$ is the empirical minimizer of $\hat{\epsilon}_\alpha(h)$ on this batch and $h_T^* = \min_{h \in \mathcal{H}} \epsilon_T(h)$ is the target error minimizer, then for any $\delta > 0$, with probability at least $1 - 2\delta$,*

$$\epsilon_T(\hat{h}) \leq \epsilon_T(h_T^*) + 4\sqrt{\frac{2d\log(2(m^r + 1)) + 2\log\left(\frac{4}{\delta}\right)}{m^r}}$$
$$+ 2(1 - \beta)\left(d_{h,\mathcal{H}}(\widehat{S}, \widehat{T}) + 2\sqrt{\frac{2d\log\frac{em'}{4d}}{m'}} + 2\sqrt{\frac{\log\frac{2}{\delta}}{2m'}} + \lambda\right), \tag{32}$$

*Proof.* According to Lemma 1, we have:

$$\epsilon_T(\hat{h}) \leq \epsilon_\alpha(\hat{h}) + (1 - \alpha)\left(d_{h,\mathcal{H}}(S, T) + \lambda\right). \tag{33}$$

By applying Lemma 3, we deduce that:

$$\epsilon_T(\hat{h}) \leq \hat{\epsilon}_\alpha(\hat{h}) + 2\sqrt{\frac{\alpha^2}{\beta} + \frac{(1-\alpha)^2}{1-\beta}} \sqrt{\frac{2d\log(2(m^r+1)) + 2\log\left(\frac{4}{\delta}\right)}{m^r}} + (1-\alpha)\left(d_{h,\mathcal{H}}(S,T) + \lambda\right)$$

$$\leq \hat{\epsilon}_\alpha\left(h_T^*\right) + 2\sqrt{\frac{\alpha^2}{\beta} + \frac{(1-\alpha)^2}{1-\beta}} \sqrt{\frac{2d\log(2(m^r+1)) + 2\log\left(\frac{4}{\delta}\right)}{m^r}} + (1-\alpha)\left(d_{h,\mathcal{H}}(S,T) + \lambda\right)$$

$$\leq \epsilon_\alpha\left(h_T^*\right) + 4\sqrt{\frac{\alpha^2}{\beta} + \frac{(1-\alpha)^2}{1-\beta}} \sqrt{\frac{2d\log(2(m^r+1)) + 2\log\left(\frac{4}{\delta}\right)}{m^r}} + (1-\alpha)\left(d_{h,\mathcal{H}}(S,T) + \lambda\right).$$

$$(34)$$

Based on Lemma 2, we obtain:

$$\epsilon_T(\hat{h}) \leq \epsilon_T\left(h_T^*\right) + 4\sqrt{\frac{\alpha^2}{\beta} + \frac{(1-\alpha)^2}{1-\beta}} \sqrt{\frac{2d\log(2(m^r+1)) + 2\log\left(\frac{4}{\delta}\right)}{m^r}}$$

$$+ 2(1-\alpha)\left(d_{h,\mathcal{H}}(\widehat{S},\widehat{T}) + 2\mathfrak{R}_{m',S}(\mathcal{H}\Delta\mathcal{H}) + \sqrt{\frac{\log\frac{2}{\delta}}{2m'}} + 2\mathfrak{R}_{m',T}(\mathcal{H}\Delta\mathcal{H}) + \sqrt{\frac{\log\frac{2}{\delta}}{2m'}} + \lambda\right).$$

$$(35)$$

According to Lemma 3 and Mohri et al. (2012), it follows that:

$$\epsilon_T(\hat{h}) \leq \epsilon_T\left(h_T^*\right) + 4\sqrt{\frac{\alpha^2}{\beta} + \frac{(1-\alpha)^2}{1-\beta}} \sqrt{\frac{2d\log(2(m^r+1)) + 2\log\left(\frac{4}{\delta}\right)}{m^r}}$$

$$+ 2(1-\alpha)\left(d_{h,\mathcal{H}}(\widehat{S},\widehat{T}) + 2\sqrt{\frac{2d\log\frac{em'}{4d}}{m'}} + 2\sqrt{\frac{\log\frac{2}{\delta}}{2m'}} + \lambda\right).$$

$$(36)$$

If we allocate the loss weight according to the number of samples, that is, setting $\alpha = \beta$, we can conclude:

$$\epsilon_T(\hat{h}) \leq \epsilon_T\left(h_T^*\right) + 4\sqrt{\frac{2d\log(2(m^r+1)) + 2\log\left(\frac{4}{\delta}\right)}{m^r}}$$

$$+ 2(1-\beta)\left(d_{h,\mathcal{H}}(\widehat{S},\widehat{T}) + 2\sqrt{\frac{2d\log\frac{em'}{4d}}{m'}} + 2\sqrt{\frac{\log\frac{2}{\delta}}{2m'}} + \lambda\right).$$

$$(37)$$

This completes the proof.

## C   PROOFS OF THEOREM 2

First, let $\tilde{R}_V(f) = \mathbb{E}\left[\widehat{R}_V(f)\right]$. Then, following the results in Maurer & Pontil (2009), which provides a uniform convergence form of Bennett's inequality (Bennett, 1962), we can derive:

$$\tilde{R}_V(f) - \widehat{R}_V(f) \leq \mathcal{O}\left(\sqrt{\mathbb{V}(f) \cdot \frac{\log\frac{\mathcal{M}_{NR}}{\delta}}{N^R}} + \frac{\log\frac{\mathcal{M}_{NR}}{\delta}}{N^R}\right) \tag{38}$$

Moreover, the following holds:

$$\left|\tilde{R}_V(f) - R_V(f)\right| = \left|\mathbb{E}\left[\widehat{R}_V(f)\right] - \mathbb{E}\left[\widehat{R}_V^\star(f)\right]\right| \leq \mathbb{E}[\|\boldsymbol{q} - \boldsymbol{p}\|_2 \cdot \sqrt{\sum_{j=1}^{c} \ell^2\left(f\left(\boldsymbol{x}_i\right), \boldsymbol{e}^{y_j}\right)}]. \tag{39}$$

Table 4: Running time of different methods on the clipart domain of the `DomainNet` dataset.

| Methods | Time (s) |
|---------|----------|
| ERM | 19.54 |
| BN | 21.03 |
| TENT | 57.23 |
| PL | 77.94 |
| SHOT-IM | 77.02 |
| T3A | 46.57 |
| TAST | 86.89 |
| TAST-BN | 128.46 |
| TSD | 105.55 |
| PROGRAM | 113.58 |
| DEYO | 92.49 |
| PASLE | 99.28 |

Table 5: Classification accuracy of PASLE under broader parameter range on the `CIFAR-10-C` dataset.

| $\tau_{start}$ | $\tau_{end}$ | Acc |
|----------------|--------------|-----|
| 0.9 | 0.8 | 77.90 |
| 0.8 | 0.7 | 77.97 |
| 0.7 | 0.6 | 77.96 |
| 0.6 | 0.5 | 77.99 |
| 0.5 | 0.4 | 77.89 |
| 0.4 | 0.3 | 77.84 |
| 0.3 | 0.2 | 77.84 |
| 0.2 | 0.1 | 77.79 |

Table 6: Classification accuracy of PASLE and its variants (PASLE-NB and PASLE-NR) on the `OfficeHome` dataset. The best performance is shown in boldface.

| Methods | A | C | P | R |
|---------|---|---|---|---|
| PASLE | **57.25±0.75** | **51.30±0.41** | **73.31±1.04** | **74.10±0.20** |
| PASLE-NB | 56.98±0.82 | 51.14±0.44 | 73.14±0.87 | 73.00±0.25 |
| PASLE-NR | 57.02±0.76 | 51.11±0.39 | 73.09±1.21 | 72.98±0.33 |

Thus, we have

$$R_V(f) - \tilde{R}(f) \leq M\sqrt{c} \cdot \mathbb{E}\left[\|\boldsymbol{q} - \boldsymbol{p}\|_2\right]. \tag{40}$$

Finally, combining Eq.38 and Eq.40 completes the proof.

## D  FURTHER ANALYSIS (APPENDIX)

**Running Time Analysis.** To evaluate the computational cost, experiments were carried out on the clipart domain of the `DomainNet` dataset, using ResNet-18 as the backbone with a batch size of 128 on an NVIDIA TITAN Xp GPU. The reported runtime excludes data loading time, ensuring fairness by using `torch.cuda.synchronize` to accurately measure the computational overhead. The results are shown in Table 4. Our method has a similar time complexity to PL in pseudo-label generation. It is a bit slower than PL because our sample utilization rate is high, and as a result, better performance has been achieved.

**Performance over a Broader Parameter Range.** We conducted a parameter sensitivity analysis experiment on the `CIFAR-10-C` dataset under shot noise with a broader range of hyperparameters. The value of $\tau_{start}$ was selected from a wider range, specifically between $0.2$ and $0.9$. The threshold gap represented as $|\tau_{start} - \tau_{end}|$, was fixed at $0.1$. For testing purposes, $\tau_{des}$ was set to $\frac{\tau_{start} - \tau_{end}}{R}$. The results are summarized in Table 5. The results indicate that the algorithm achieves optimal performance when $\tau$ is within the range of $0.5$ to $0.8$. Within a reasonable range of $\tau$, the algorithm also delivers comparable results. However, when $\tau$ is set too low (e.g., within the range of $0.1$ to $0.2$), many samples with incorrect supervision are introduced, leading to a decline in performance.

**Ablation Study of Modules in PASLE.** We additionally conducted ablation studies using two simplified variants of our framework: PASLE-NB and PASLE-NR. In PASLE-NB, the buffer is removed from the framework, while in PASLE-NR, the strategy of threshold reduction is excluded. For this study, we utilized the `OfficeHome` dataset and employed ResNet-18 as the backbone. The results

Table 7: Classification accuracy of PASLE with different candidate label selection strategies on the OfficeHome dataset. The best performance is shown in boldface.

| Methods | A | C | P | R |
|---|---|---|---|---|
| PASLE | **57.25±0.75** | **51.30±0.41** | **73.31±1.04** | **74.10±0.20** |
| PASLE-TB | 57.07±0.63 | 51.13±0.45 | 73.11±0.94 | 73.92±0.34 |
| PASLE-KB | 56.31±0.59 | 50.69±0.35 | 72.82±0.69 | 73.56±0.21 |

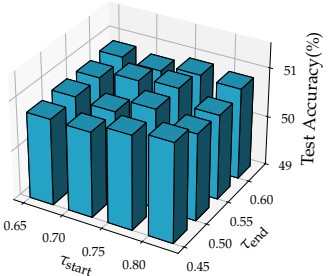 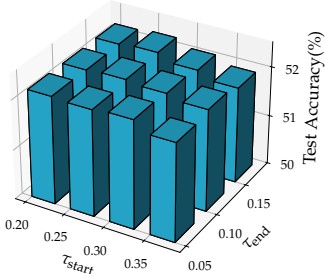

(a) The sensitivity of $\tau_{\text{start}}$ and $\tau_{\text{end}}$ on OfficeHome.   (b) The sensitivity of $\tau_{\text{start}}$ and $\tau_{\text{end}}$ on DomainNet.

Figure 3: The parameter sensitivity of $\tau_{\text{start}}$ and $\tau_{\text{end}}$ for PASLE.

are shown in Table 6, highlighting that the buffer mechanism and threshold reduction strategy, as pluggable modules in our framework, further improve its performance.

**Different Candidate Labels Generation Methods.** We further explored two approaches for generating candidate labels. The first is a threshold-based approach (PASLE-TB), where a threshold is set, and all classes with prediction probabilities exceeding this threshold are selected as candidate labels. This method generates both one-hot pseudo-labels and candidate pseudo-label sets. The second approach is top-$k$ based (PASLE-KB), where prediction probabilities are sorted in descending order, and the top-$k$ classes are chosen as candidate labels. Unlike the first method, this approach only produces candidate pseudo-label sets. The threshold and $k$ are dynamically adjusted during the adaptation process. Experiments were conducted on the OfficeHome dataset using ResNet-18 as the backbone, and the results are presented in Table 7. It can be observed that PASLE-TB achieves performance comparable to PASLE, while PASLE-KB, which lacks sample selection and directly uses the top-$k$ predicted classes of all samples as candidate labels, performs significantly worse than PASLE.

**More Parameter Sensitivity Analysis Results** We additionally conducted sensitivity analyses on $\tau_{start}$ and $\tau_{end}$ using the clipart domain of the OfficeHome dataset and the clipart domain of the DomainNet dataset as target domains. We also compared the performance of our method and the baseline methods under varying batch sizes, using the sketch domain of DomainNet and the shot noise corruption in CIFAR-100-C as target domains. The results are shown in Figure 3 and Figure 4. It can be observed that PASLE is robust to hyperparameter selection and consistently outperforms other methods across different batch sizes.

## E   FULL EXPERIMENTAL RESULTS

Table 8: Full results on the `PACS` dataset.

| Methods | BackBone | A | C | P | S | Avg. |
|---------|----------|---|---|---|---|------|
| ERM | | 78.73±1.69 | 74.84±0.54 | 95.03±0.31 | 68.87±3.36 | 79.37 |
| BN | | 82.03±0.55 | 81.09±0.45 | 96.08±0.33 | 73.12±0.44 | 83.08 |
| TENT | | 82.19±0.59 | 81.27±0.47 | 95.85±0.33 | 73.62±0.68 | 83.23 |
| PL | | 85.29±1.19 | 83.72±0.97 | 95.99±0.42 | 77.65±2.51 | 85.66 |
| SHOT-IM | | 84.75±0.66 | 81.83±1.27 | 96.33±0.78 | 69.18±0.55 | 83.02 |
| T3A | ResNet-18 | 80.75±1.45 | 78.16±0.61 | 95.89±0.55 | 72.01±2.99 | 81.70 |
| TAST | | 83.32±0.88 | 82.44±0.92 | 96.27±0.54 | 75.19±0.89 | 84.31 |
| TAST-BN | | 87.09±0.32 | 83.89±1.37 | 96.69±0.39 | 77.74±1.45 | 86.35 |
| TSD | | 87.81±0.64 | 87.09±0.50 | 96.71±0.57 | 79.89±0.32 | 87.88 |
| PROGRAM | | 84.72±0.55 | 80.36±0.60 | 96.05±0.39 | 73.14±0.40 | 83.57 |
| DEYO | | 85.84±1.10 | 83.39±0.56 | 96.13±0.48 | 79.67±0.80 | 86.26 |
| PASLE | | 88.19±1.42 | 87.09±0.24 | 96.83±0.48 | 80.51±1.28 | 88.16 |
| ERM | | 85.84±0.78 | 79.78±2.15 | 96.47±0.37 | 81.27±2.12 | 85.84 |
| BN | | 86.73±0.82 | 84.13±1.98 | 96.79±0.07 | 76.36±1.53 | 86.00 |
| TENT | | 87.00±0.92 | 84.51±1.89 | 96.89±0.10 | 77.65±1.53 | 86.51 |
| PL | | 88.80±0.88 | 82.94±4.90 | 95.35±2.32 | 75.53±5.04 | 85.66 |
| SHOT-IM | | 85.68±1.56 | 83.52±0.47 | 95.05±0.63 | 76.83±1.93 | 85.27 |
| T3A | ResNet-50 | 86.51±0.41 | 81.67±1.64 | 96.85±0.18 | 81.12±2.03 | 86.54 |
| TAST | | 87.84±0.53 | 84.56±2.21 | 97.35±0.19 | 78.00±1.13 | 86.94 |
| TAST-BN | | 89.94±0.31 | 86.68±1.01 | 97.49±0.48 | 83.77±1.77 | 89.47 |
| TSD | | 91.06±0.56 | 90.67±0.68 | 97.70±0.15 | 85.09±1.21 | 91.13 |
| PROGRAM | | 87.21±1.30 | 84.09±1.86 | 96.89±0.06 | 76.44±1.64 | 86.16 |
| DEYO | | 88.36±0.94 | 85.24±1.61 | 97.05±0.12 | 82.26±0.54 | 88.23 |
| PASLE | | 91.57±0.48 | 89.88±1.53 | 97.74±0.28 | 86.25±0.78 | 91.36 |

Table 9: Full results on the `VLCS` dataset.

| Methods | BackBone | V | L | C | S | Avg. |
|---------|----------|---|---|---|---|------|
| ERM | | 95.66±1.31 | 63.09±1.59 | 69.17±0.88 | 72.62±3.70 | 75.14 |
| BN | | 82.71±2.46 | 58.83±1.55 | 62.20±0.86 | 71.42±2.13 | 68.79 |
| TENT | | 83.30±2.36 | 59.26±1.57 | 62.77±1.02 | 71.80±1.91 | 69.28 |
| PL | | 92.32±1.63 | 63.87±1.41 | 69.47±0.97 | 73.05±1.42 | 74.68 |
| SHOT-IM | | 88.06±3.50 | 58.58±1.62 | 63.50±1.94 | 73.06±2.02 | 70.80 |
| T3A | ResNet-18 | 98.49±1.27 | 64.02±1.60 | 68.90±0.82 | 71.92±4.00 | 75.83 |
| TAST | | 94.74±2.04 | 56.63±2.07 | 63.97±0.62 | 71.41±2.61 | 71.69 |
| TAST-BN | | 97.34±0.84 | 65.02±1.57 | 65.71±0.87 | 72.61±5.40 | 75.17 |
| TSD | | 96.30±0.47 | 65.47±0.14 | 67.84±0.53 | 72.25±2.88 | 75.47 |
| PROGRAM | | 95.87±1.45 | 58.71±0.81 | 60.12±1.43 | 71.85±2.47 | 71.64 |
| DEYO | | 95.16±1.35 | 63.93±0.37 | 67.20±0.97 | 73.33±1.80 | 74.91 |
| PASLE | | 96.04±1.53 | 66.48±0.51 | 72.64±0.45 | 76.48±0.29 | 77.91 |
| ERM | | 97.15±0.43 | 63.08±0.65 | 70.63±0.76 | 73.37±1.14 | 76.06 |
| BN | | 77.90±4.52 | 56.66±2.09 | 63.36±1.14 | 73.11±0.37 | 67.76 |
| TENT | | 79.15±4.67 | 57.05±2.06 | 63.92±1.09 | 73.50±0.18 | 68.41 |
| PL | | 91.31±3.18 | 61.60±3.90 | 70.97±0.05 | 71.32±2.66 | 73.80 |
| SHOT-IM | | 80.60±3.95 | 55.78±2.63 | 63.51±1.94 | 74.06±0.73 | 68.49 |
| T3A | ResNet-50 | 98.18±0.11 | 64.13±1.10 | 72.04±2.37 | 71.99±0.81 | 76.59 |
| TAST | | 81.70±4.20 | 52.10±0.42 | 62.29±1.45 | 73.20±1.04 | 67.32 |
| TAST-BN | | 96.77±1.42 | 61.33±0.06 | 68.96±4.14 | 75.30±0.19 | 75.59 |
| TSD | | 93.92±1.28 | 58.03±1.04 | 69.65±2.52 | 77.49±0.99 | 74.77 |
| PROGRAM | | 85.05±4.15 | 58.42±0.40 | 60.12±1.43 | 71.82±1.50 | 68.85 |
| DEYO | | 84.38±1.70 | 60.92±1.97 | 67.20±0.97 | 73.85±0.21 | 71.59 |
| PASLE | | 96.01±0.89 | 66.20±1.98 | 76.01±0.91 | 76.59±0.88 | 78.70 |

Table 10: Full results on the `OfficeHome` dataset.

| Methods | BackBone | A | C | P | R | Avg. |
|---------|----------|---|---|---|---|------|
| ERM | | 55.76±0.80 | 48.66±0.43 | 71.46±0.67 | 73.84±0.23 | 62.43 |
| BN | | 54.96±0.58 | 49.74±0.37 | 70.95±0.91 | 73.50±0.43 | 62.29 |
| TENT | | 55.13±0.65 | 50.00±0.21 | 71.27±0.89 | 73.63±0.43 | 62.51 |
| PL | | 55.09±0.36 | 50.74±0.19 | 71.17±1.25 | 73.83±0.39 | 62.71 |
| SHOT-IM | | 56.75±1.12 | 52.03±0.57 | 72.77±0.05 | 74.10±0.33 | 63.91 |
| T3A | ResNet-18 | 56.52±1.12 | 50.62±0.67 | 73.45±0.71 | 74.99±0.33 | 63.90 |
| TAST | | 55.75±0.89 | 51.71±0.49 | 73.93±0.95 | 74.46±0.34 | 63.96 |
| TAST-BN | | 55.28±0.65 | 50.49±0.64 | 71.89±1.46 | 72.07±0.44 | 62.43 |
| TSD | | 57.27±0.71 | 50.60±1.59 | 72.24±0.85 | 73.55±0.21 | 63.42 |
| PROGRAM | | 56.69±1.45 | 50.94±0.51 | 71.80±0.40 | 73.98±0.47 | 63.35 |
| DEYO | | 56.50±0.30 | 50.80±0.15 | 71.97±1.02 | 73.92±0.41 | 63.30 |
| PASLE | | 57.25±0.75 | 51.30±0.41 | 73.31±1.04 | 74.10±0.20 | 63.99 |
| ERM | | 63.04±0.48 | 53.88±0.18 | 76.55±0.41 | 77.89±0.19 | 67.84 |
| BN | | 62.00±0.81 | 53.45±0.47 | 75.08±0.66 | 76.76±0.74 | 66.82 |
| TENT | | 62.44±0.72 | 54.11±0.48 | 75.68±0.67 | 76.84±0.57 | 67.27 |
| PL | | 63.32±0.39 | 54.73±0.99 | 73.89±1.22 | 77.30±0.46 | 67.31 |
| SHOT-IM | | 62.95±1.15 | 54.56±0.44 | 76.37±0.95 | 77.66±0.29 | 67.89 |
| T3A | ResNet-50 | 63.26±0.23 | 55.30±0.32 | 78.13±0.44 | 78.72±0.45 | 68.85 |
| TAST | | 63.42±0.59 | 55.61±0.59 | 78.01±0.81 | 77.74±0.46 | 68.70 |
| TAST-BN | | 62.76±0.62 | 55.01±0.43 | 77.01±0.85 | 77.11±0.74 | 67.97 |
| TSD | | 64.32±0.41 | 56.91±0.97 | 77.19±0.67 | 77.46±0.45 | 68.97 |
| PROGRAM | | 63.36±0.87 | 54.27±0.23 | 77.24±0.69 | 77.24±0.49 | 68.03 |
| DEYO | | 63.77±0.23 | 54.90±1.10 | 76.36±0.51 | 77.28±0.56 | 68.08 |
| PASLE | | 65.47±0.83 | 56.08±0.82 | 78.11±0.65 | 77.83±0.30 | 69.37 |

Table 11: Full results on the `DomainNet` dataset.

| Methods | BackBone | clipart | infograph | painting | quickdraw | real | sketch | Avg |
|---------|----------|---------|-----------|----------|-----------|------|--------|-----|
| ERM | | 50.48±0.21 | 15.31±0.15 | 41.86±0.12 | 11.66±0.49 | 51.71±0.29 | 43.53±0.02 | 35.76 |
| BN | | 50.74±0.09 | 11.38±0.03 | 40.76±0.12 | 11.29±0.39 | 51.78±0.19 | 43.72±0.26 | 34.95 |
| TENT | | 51.13±0.07 | 12.52±0.19 | 41.91±0.14 | 10.57±0.38 | 51.31±0.25 | 44.76±0.17 | 35.37 |
| PL | | 50.88±0.06 | 12.87±0.70 | 41.22±0.10 | 10.84±0.74 | 51.62±0.29 | 44.01±0.25 | 35.24 |
| SHOT-IM | | 51.02±0.07 | 12.75±0.39 | 41.41±0.13 | 13.71±0.32 | 52.26±0.15 | 44.38±0.23 | 35.92 |
| T3A | ResNet-18 | 50.34±0.26 | 15.11±0.20 | 40.35±0.12 | 16.24±0.23 | 53.13±0.30 | 42.68±0.17 | 36.31 |
| TAST | | 50.31±0.16 | 12.64±0.09 | 40.66±0.05 | 14.59±0.51 | 53.58±0.15 | 42.49±0.26 | 35.71 |
| TAST-BN | | 50.44±0.27 | 13.21±0.13 | 40.97±0.12 | 14.72±0.49 | 52.39±0.41 | 43.19±0.28 | 35.82 |
| TSD | | 50.74±0.11 | 13.58±0.07 | 42.93±0.24 | 11.78±0.34 | 51.95±0.21 | 44.20±0.20 | 35.86 |
| PROGRAM | | 51.07±0.16 | 13.27±0.26 | 41.52±0.12 | 13.40±0.41 | 52.31±0.28 | 44.27±0.25 | 35.97 |
| DEYO | | 50.86±0.05 | 13.19±0.29 | 41.23±0.12 | 11.19±0.50 | 51.81±0.21 | 43.96±0.23 | 35.37 |
| PASLE | | 51.76±0.39 | 14.98±0.28 | 43.06±0.15 | 13.67±0.26 | 52.69±0.18 | 45.15±0.27 | 36.89 |
| ERM | | 61.00±0.24 | 20.81±0.19 | 49.58±0.06 | 13.57±0.26 | 61.95±0.16 | 52.07±0.36 | 43.16 |
| BN | | 60.58±0.23 | 15.11±0.17 | 48.64±0.08 | 11.92±0.20 | 61.06±0.16 | 51.70±0.16 | 41.50 |
| TENT | | 61.64±0.15 | 17.41±0.01 | 50.40±0.17 | 10.11±0.58 | 61.40±0.14 | 53.30±0.08 | 42.38 |
| PL | | 61.04±0.22 | 17.90±0.20 | 49.90±0.06 | 11.57±0.15 | 61.26±0.04 | 52.61±0.19 | 42.38 |
| SHOT-IM | | 61.29±0.21 | 17.56±0.14 | 49.81±0.10 | 16.50±0.52 | 62.52±0.10 | 52.79±0.22 | 43.41 |
| T3A | ResNet-50 | 61.05±0.23 | 20.94±0.14 | 48.71±0.08 | 18.55±0.39 | 63.19±0.08 | 51.57±0.27 | 44.00 |
| TAST | | 60.65±0.22 | 17.93±0.19 | 49.03±0.19 | 15.17±0.28 | 62.62±0.12 | 51.64±0.07 | 42.84 |
| TAST-BN | | 61.03±0.32 | 18.04±0.22 | 49.65±0.19 | 14.75±0.41 | 62.71±0.07 | 51.97±0.09 | 43.03 |
| TSD | | 60.76±0.24 | 17.89±0.17 | 49.82±0.65 | 12.21±0.26 | 61.66±0.17 | 52.27±0.18 | 42.44 |
| PROGRAM | | 61.15±0.26 | 18.08±0.04 | 49.85±0.06 | 15.53±0.40 | 62.14±0.04 | 53.30±0.20 | 43.34 |
| DEYO | | 61.04±0.22 | 18.17±0.11 | 49.84±0.05 | 11.99±0.23 | 61.25±0.02 | 52.50±0.21 | 42.47 |
| PASLE | | 62.46±0.39 | 20.67±0.12 | 51.74±0.16 | 16.73±0.51 | 63.76±0.09 | 54.11±0.26 | 44.91 |

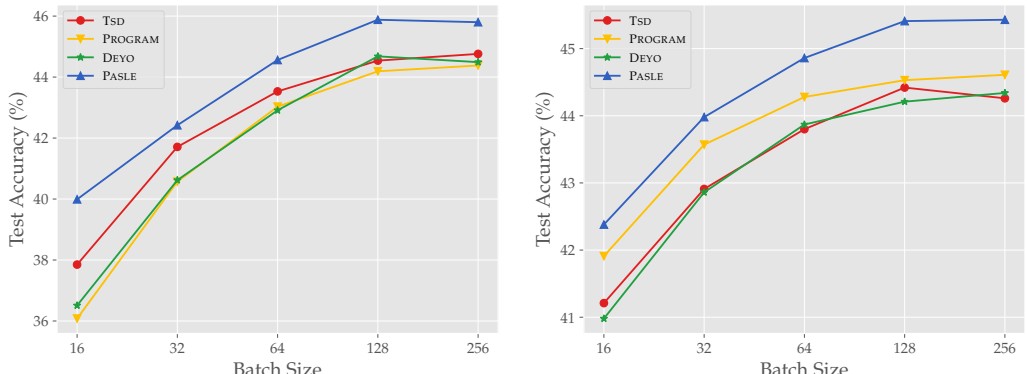

(a) The sensitivity of batch size on `CIFAR-100-C`.    (b) The sensitivity of batch size on `DomainNet`.

Figure 4: The parameter sensitivity of batch size for PASLE.

Table 12: Full results on the `CIFAR-10-C` dataset.

| Methods | shot | motn | snow | pixel | gauss | defoc | brit | fog | zoom | frost | glass | impul | contr | jpeg | elast | Avg. |
|---|---|---|---|---|---|---|---|---|---|---|---|---|---|---|---|---|
| ERM | 24.93 | 19.40 | 26.12 | 21.62 | 24.52 | 19.59 | 28.41 | 13.48 | 19.53 | 19.13 | 18.31 | 18.34 | 10.93 | 24.93 | 20.59 | 20.66 |
| BN | 76.79 | 75.93 | 78.17 | 80.13 | 77.33 | 79.13 | 83.08 | 70.64 | 79.25 | 76.05 | 68.49 | 65.80 | 62.82 | 80.16 | 76.16 | 75.33 |
| TENT | 76.80 | 76.11 | 78.23 | 80.07 | 77.29 | 79.12 | 83.13 | 70.80 | 79.31 | 76.21 | 68.57 | 66.03 | 63.19 | 80.24 | 76.04 | 75.41 |
| PL | 77.07 | 76.45 | 78.25 | 80.42 | 77.59 | 79.56 | 83.20 | 71.26 | 79.77 | 76.50 | 68.74 | 66.22 | 63.95 | 80.21 | 76.34 | 75.70 |
| SHOT-IM | 77.20 | 76.67 | 78.43 | 80.54 | 77.63 | 79.77 | 83.35 | 71.35 | 79.90 | 76.42 | 68.85 | 66.60 | 64.46 | 80.24 | 76.30 | 75.85 |
| T3A | 29.97 | 20.27 | 29.93 | 26.04 | 29.30 | 19.81 | 34.00 | 13.64 | 20.02 | 21.03 | 21.28 | 21.91 | 15.94 | 28.33 | 21.29 | 23.52 |
| TAST | 75.27 | 75.16 | 77.07 | 78.99 | 76.02 | 78.05 | 82.10 | 69.39 | 78.01 | 75.47 | 66.74 | 64.42 | 62.55 | 78.11 | 74.64 | 74.13 |
| TAST-BN | 75.82 | 74.84 | 77.28 | 79.35 | 76.56 | 78.29 | 83.30 | 69.34 | 78.64 | 75.17 | 67.74 | 64.73 | 62.17 | 79.45 | 75.69 | 74.56 |
| TSD | 76.54 | 75.58 | 78.28 | 79.96 | 77.18 | 78.79 | 83.16 | 70.32 | 78.84 | 76.03 | 68.01 | 66.06 | 62.10 | 80.09 | 76.18 | 75.14 |
| PROGRAM | 76.73 | 76.22 | 77.87 | 79.98 | 77.10 | 78.83 | 83.21 | 69.86 | 78.74 | 76.01 | 67.88 | 65.40 | 61.21 | 79.95 | 76.06 | 75.00 |
| DEYO | 77.02 | 76.89 | 78.41 | 80.52 | 77.42 | 79.79 | 83.25 | 71.28 | 79.90 | 76.41 | 68.50 | 66.18 | 64.04 | 80.36 | 76.20 | 75.74 |
| PASLE | 78.03 | 77.40 | 79.08 | 81.16 | 78.22 | 80.81 | 83.83 | 72.31 | 80.81 | 77.28 | 69.34 | 67.07 | 67.17 | 80.75 | 76.77 | 76.67 |

Table 13: Full results on the `CIFAR-100-C` dataset.

| Methods | shot | motn | snow | pixel | gauss | defoc | brit | fog | zoom | frost | glass | impul | contr | jpeg | elast | Avg. |
|---|---|---|---|---|---|---|---|---|---|---|---|---|---|---|---|---|
| ERM | 7.88 | 3.86 | 8.30 | 6.95 | 7.95 | 4.07 | 11.02 | 1.63 | 3.95 | 5.89 | 5.13 | 5.45 | 1.08 | 8.54 | 5.89 | 5.84 |
| BN | 44.17 | 46.23 | 44.98 | 50.01 | 44.27 | 48.36 | 50.36 | 37.55 | 48.89 | 44.02 | 38.78 | 34.57 | 31.69 | 48.78 | 45.57 | 43.88 |
| TENT | 44.19 | 46.23 | 45.02 | 49.97 | 44.34 | 48.55 | 50.47 | 37.53 | 48.89 | 44.07 | 38.81 | 34.57 | 31.68 | 48.91 | 45.65 | 43.93 |
| PL | 44.57 | 47.05 | 45.09 | 50.11 | 44.36 | 49.27 | 50.58 | 37.32 | 49.49 | 44.35 | 39.29 | 34.94 | 32.22 | 49.01 | 45.92 | 44.24 |
| SHOT-IM | 44.51 | 47.23 | 45.20 | 50.34 | 44.81 | 49.22 | 50.76 | 37.93 | 49.25 | 44.33 | 39.16 | 34.90 | 33.05 | 49.04 | 45.60 | 44.36 |
| T3A | 8.24 | 5.28 | 9.14 | 8.19 | 8.59 | 5.04 | 12.94 | 2.23 | 5.50 | 6.18 | 6.23 | 5.97 | 1.08 | 9.49 | 6.98 | 6.74 |
| TAST | 39.65 | 42.29 | 40.35 | 44.83 | 39.86 | 43.72 | 45.00 | 33.13 | 43.37 | 39.13 | 35.68 | 31.45 | 25.91 | 43.23 | 40.50 | 39.21 |
| TAST-BN | 32.75 | 33.92 | 32.96 | 35.82 | 32.85 | 35.32 | 37.34 | 27.15 | 35.44 | 33.10 | 27.77 | 24.79 | 20.60 | 35.59 | 32.22 | 31.84 |
| TSD | 44.54 | 46.55 | 45.32 | 50.47 | 44.76 | 48.92 | 50.97 | 37.38 | 49.46 | 44.25 | 39.05 | 34.62 | 31.36 | 49.26 | 45.95 | 44.19 |
| PROGRAM | 44.19 | 46.63 | 44.97 | 49.91 | 44.63 | 48.76 | 50.36 | 37.76 | 48.74 | 44.37 | 38.87 | 34.78 | 32.42 | 48.78 | 45.67 | 44.06 |
| DEYO | 44.68 | 47.21 | 45.00 | 50.19 | 44.56 | 49.06 | 50.57 | 37.88 | 49.39 | 44.41 | 39.36 | 34.98 | 32.13 | 48.92 | 45.82 | 44.28 |
| PASLE | 45.88 | 48.44 | 46.28 | 51.43 | 45.60 | 50.05 | 51.92 | 39.12 | 50.42 | 45.17 | 39.94 | 36.00 | 32.87 | 50.05 | 46.63 | 45.32 |

