# OpenReview forum: "Selective Label Enhancement Learning for Test-Time Adaptation"
_ICLR.cc/2025/Conference — ICLR 2025 Poster_

### Official Review · Reviewer_b8X9 · 2024-10-26

**Soundness:** 3
**Presentation:** 3
**Contribution:** 2
**Rating:** 6
**Confidence:** 3

**Summary:**

The paper introduces the Progressive Adaptation with Selective Label Enhancement (PASLE) framework for test-time adaptation (TTA). Unlike traditional methods that assign definite pseudo-labels, PASLE assigns candidate pseudo-label sets to uncertain test samples while providing one-hot labels to confident samples. This approach allows the model to adapt progressively, refining the uncertain pseudo-labels based on the model's evolving understanding of the target domain.

**Strengths:**

1. PASLE effectively partitions test samples into confident and uncertain subsets, improving labeling accuracy for uncertain samples.

2. The model is trained iteratively on both certain and uncertain pseudo-labeled data, enhancing adaptation capabilities over time.

3. The paper establishes a generalization bound that suggests increased supervision from target domain samples can lead to improved model performance.

**Weaknesses:**

1. I am confused about some notations in theorem 1, what is the specific meaning of d_{h, H}(S,  T)? It seems that d_{h, H}(S,  T) is a constant with your algorithm. Does theorem 1 show the superiority of your algorithm because there is always a constant gap between the \epsilon_T(\hat{h}) and \epsilon_T (h_T^*)?

2. Whether using the two hyper-parameter to control the iteration is reasonable in equation 9? Since different datasets have different parameters and there is no prior knowledge to guide us in choosing suitable parameters, making hard to achieve the best results

3. More experiments about the sensitivity of the τ_start, τ_end, and batch size on other datasets are expected to be seen.

**Questions:**

See weakness

---

> ### Author Response · Authors · 2024-11-24
> **Response to Reviewer b8X9 (1/2)**
>
> Thank you for taking the time to carefully review our paper and offer valuable feedback. In response to your concerns, we would like to provide the following explanations.
>
> 1. **Q1:** I am confused about some notations in theorem 1, what is the specific meaning of $d_{h, \mathcal{H}}(S, T)$? It seems that $d_{h, \mathcal{H}}(S, T)$ is a constant with your algorithm. Does theorem 1 show the superiority of your algorithm because there is always a constant gap between the $\epsilon_T(\hat{h})$ and $\epsilon_T (h_T^*)$?
>
>    **A1:** $d_{h, \mathcal{H}}(S, T)$ is a proper statistic to measure the distribution shift from the source domain $S$ to target domain $T$ [1]. For a specific classifier $h$, $d_{h, \mathcal{H}}(S, T)$ is a constant that quantifies the discrepancy between $S$ and $T$, under the assumption that the hypothesis space $\mathcal{H}$, the source domain and the target domain remain fixed. In domain adaptation theory, the domain discrepancy term is fundamental and has been widely utilized in guiding the design of various methods [2-4].
>
>    Our method aims to tighten the generalization error bound by introducing candidate labels to offer effective supervision for uncertain samples and a buffer that temporarily stores currently unusable samples to provide effective supervision in the future, thereby enabling the utilization of more target domain samples with effective supervision. As analyzed in the article, the growth rate of the first term in the generalization error bound is $\mathcal{O}(\sqrt{\frac{\log m}{m}})$, it decreases as more samples are incorporated into adaptation. Meanwhile, the coefficient of the second term, $1-\beta$, also decreases as the proportion of target domain samples relative to the total number of samples increases. Consequently, the overall generalization error bound becomes tighter as the number of target domain samples with effective supervision increases, which demonstrates the advantage of our algorithm.
>
> 2. **Q2:** Whether using the two hyper-parameter to control the iteration is reasonable in equation 9? Since different datasets have different parameters and there is no prior knowledge to guide us in choosing suitable parameters, making hard to achieve the best results
>
>    **A2:** Using Equation (9) to control the linear decay of the threshold is one approach for threshold iteration. In both the paper and subsequent sensitivity analyses on additional datasets (Table 1, Table 2), our method consistently demonstrates robust performance under various settings of $\tau_{start}$, $\tau_{end}$, and $\tau_{des}$ when using the threshold decay approach from Equation (9). This indicates that the algorithm can achieve strong performance across different datasets without requiring excessive hyperparameter tuning, as long as the chosen hyperparameters fall within a reasonably large range guided by the number of classes. Moreover, the conservative initial threshold setting, the incorporation of uncertainty, and the buffer’s temporary storage mechanism ensure the reliability of sample supervision under diverse hyperparameter choices. These factors collectively contribute to the significant performance improvements observed in our experiments.

---

> ### Author Response · Authors · 2024-11-24
> **Response to Reviewer b8X9 (2/2)**
>
> 3. **Q3:** More experiments about the sensitivity of the $\tau_{start}$, $\tau_{end}$, and batch size on other datasets are expected to be seen.
>
>    **A3:** We additionally conducted sensitivity analyses on $\tau_{start}$ and $\tau_{end}$ using the Clipart domain of the OfficeHome dataset and the Clipart domain of the DomainNet dataset as target domains. For testing purposes,  $\tau_{des}$ was set as $\frac{\tau_\text{start} - \tau_\text{end}}{R}$. The results are summarized separately in the tables below:
>
>    **Table 1:** The sensitivity of $\tau_{start}$ and $\tau_{end}$ on OfficeHome dataset.
>
>    | $\tau_{start},\tau_{end}$ | 0.8   | 0.75  | 0.7   | 0.65  |
>    | ------------------------- | ----- | ----- | ----- | ----- |
>    | 0.6                       | 50.84 | 50.90 | 50.77 | 50.84 |
>    | 0.55                      | 50.71 | 51.02 | 50.95 | 50.79 |
>    | 0.5                       | 50.75 | 51.00 | 50.72 | 50.79 |
>    | 0.45                      | 51.00 | 50.90 | 50.72 | 50.74 |
>
>    **Table 2:** The sensitivity of $\tau_{start}$ and $\tau_{end}$ on DomainNet dataset.
>
>    | $\tau_{start},\tau_{end}$ | 0.35  | 0.3   | 0.25  | 0.2   |
>    | ------------------------- | ----- | ----- | ----- | ----- |
>    | 0.15                      | 51.93 | 52.07 | 52.21 | 52.17 |
>    | 0.1                       | 52.05 | 52.14 | 52.19 | 52.16 |
>    | 0.05                      | 51.98 | 52.18 | 52.14 | 52.17 |
>
>    The results indicate that the performance of PASLE remains relatively stable across a wide range of values for each hyperparameter. We also compared the performance of our method and the baseline methods under varying batch sizes, using the sketch domain of DomainNet and the shot noise corruption in CIFAR-100-C as target domains. The results are presented separately in the tables below:
>
>    **Table 3:** The sensitivity of batch size on DomainNet dataset.
>
>    | Batch Size | 16    | 32    | 64    | 128   | 256   |
>    | ---------- | ----- | ----- | ----- | ----- | ----- |
>    | TSD        | 41.21 | 42.91 | 43.80 | 44.42 | 44.26 |
>    | PROGRAM    | 41.91 | 43.57 | 44.28 | 44.53 | 44.61 |
>    | DeYO       | 40.98 | 42.86 | 43.87 | 44.21 | 44.34 |
>    | PASLE      | 42.38 | 43.98 | 44.86 | 45.41 | 45.43 |
>
>    **Table 4:** The sensitivity of batch size on CIFAR-100-C dataset.
>
>    | Batch Size | 16    | 32    | 64    | 128   | 256   |
>    | ---------- | ----- | ----- | ----- | ----- | ----- |
>    | TSD        | 37.85 | 41.71 | 43.53 | 44.54 | 44.76 |
>    | PROGRAM    | 36.08 | 40.57 | 43.03 | 44.19 | 44.38 |
>    | DeYO       | 36.51 | 40.62 | 42.91 | 44.68 | 44.49 |
>    | PASLE      | 39.99 | 42.42 | 44.56 | 45.88 | 45.60 |
>
>    As shown, our method consistently outperforms the other methods across different batch sizes.
>
> **Refs.**
>
> [1] On Localized Discrepancy for Domain Adaptation. arXiv:2008.06242
>
> [2] Gradient Distribution Alignment Certificates Better Adversarial Domain Adaptation. ICCV 2021
>
> [3] A Closer Look at Smoothness in Domain Adversarial Training. ICML 2022
>
> [4] MADG: Margin-based Adversarial Learning for Domain Generalization. NeurIPS 2023

---

> > ### Comment · Reviewer_b8X9 · 2024-11-27
> >
> > Thanks for the author's response. I still have some doubts about question 1. If there is a constant in the generalization bound of your theorem, does that mean that even if your model gets an infinite number of samples, it will not achieve optimal performance? This seems to go against common sense, please give me a reasonable answer.
> >
> > Thanks for the author's reply to questions 2 and 3, I have no further questions about these two questions.

---

> > > ### Author Response · Authors · 2024-11-27
> > >
> > > Thank you for your reply! Next, we will further explain Q1.
> > >
> > > In A1, we mention that 'For a specific classifier $h$, $d_{h, \mathcal{H}}(S, T)$ is a constant that quantifies the discrepancy between $S$ and $T$ ... ', which means that if the classifier $h$ satisfies certain optimization conditions, $d_{h, \mathcal{H}}(S, T)$ will tend to approach 0. In fact, this term $d_{h, \mathcal{H}}(S, T)$ motivates many approaches [1-2] in domain adaptation to pursue the minimization of the discrepancy.
> > >
> > > Besides,  in our theorem, the discrepancy $d_{h, \mathcal{H}}(S, T)$ and its coefficient $(1-\beta)$ are incorporated as a trade-off to  control the generalization bound.  Let us consider two special cases: (1) the source domain $S$ and the target domain $T$ completely overlap, and (2) $S$ and $T$ are entirely disjoint.
> > >
> > > In the first case, where $S$ and $T$ completely overlap, there is no difference between the two domains, resulting in $d_{h, \mathcal{H}}(S, T) = 0$. Consequently, the generalization error bound does not include this constant term.
> > >
> > > In the second case, where $S$ and $T$ do not overlap at all, the test data stream contains no samples from $S$, meaning $1-\beta = 0$. In this scenario, the generalization error bound depends only on the first term, whose growth rate is $\mathcal{O}(\sqrt{\frac{\log m}{m}})$. If the model has access to an infinite number of samples, it will achieve optimal performance.
> > >
> > > **Refs.**
> > >
> > > [1] Bridging Theory and Algorithm for Domain Adaptation. ICML 2019
> > >
> > > [2] Gradient Distribution Alignment Certificates Better Adversarial Domain Adaptation. ICCV 2021

---

> > > > ### Comment · Reviewer_b8X9 · 2024-11-28
> > > >
> > > > Thanks for the author's reply, which solved most of my doubts, and I will raise my score

---

### Official Review · Reviewer_LuhJ · 2024-10-30

**Soundness:** 2
**Presentation:** 2
**Contribution:** 2
**Rating:** 6
**Confidence:** 3

**Summary:**

This article proposes a method for addressing the TTA problem by dividing confident samples from uncertain samples and progressively updating pseudo-labels, alleviating errors caused by unconfident pseudo-labels in TTA scenarios. The article provides a systematic and comprehensive theoretical generalization error bound and validates its effectiveness on multiple benchmark datasets.

**Strengths:**

TTA is a critical research area for machine learning models to adapt to distribution shifts in real-world scenarios, particularly with wide applications in fields such as autonomous driving and medical image analysis. This article enhances model performance on TTA issues by dynamically adjusting pseudo-labels and capturing their uncertainties. Additionally, the detailed derivation of the generalization error bound in the article offers theoretical guarantees.

**Weaknesses:**

The paper lacks sufficient persuasive experimental evidence, suggesting the need to include relevant ablation studies, discussions on time overhead, and the rationale behind experimental settings. The novelty of the paper is not distinctly highlighted, requiring a more detailed discussion of the differences from related methods. The theoretical guidance is relatively weak; exploring the quantification of pseudo-label errors could strengthen the paper.

**Questions:**

The main concerns are as follows:
1. The novelty of this paper should be emphasized more clearly. From the motivation perspective, dividing confident and non-confident samples and applying progressive training is a fairly conventional approach. Similar ideas have been extensively used in domain adaptation (DA) problems, and several papers in TTA focus on pseudo-labeling. The authors should pay more attention to these closely related works to highlight the novelty of this paper better.
2. The generalization error bound provided is a little general and offers limited guidance for the current problem. Based on the motivation of the paper, if the so-called more effective supervised information can be quantified? If pseudo-label error terms or confidence levels could be incorporated, it would help reveal how the label-generation process impacts generalization performance, thereby offering more practical insights. Additionally, how is the divergence term in the bound reduced in this paper? How does it influence pseudo-labeling and progressive adaptation?
3. Regarding the experimental setup, the datasets used in this paper differ from those employed in previous methods. The rationale for these choices should be explained in detail. Furthermore, for certain methods with the same settings like PROGRAM, why do the results differ from the original paper when using the same benchmark and backbone? Could it be due to different settings or other reasons? This should be clarified in the paper, as such vague experimental setups and comparisons make it difficult for readers to accurately assess the actual performance of the method.
4. The paper lacks ablation studies to evaluate the effectiveness of each module. Additionally, since the proposed method is an online model, time efficiency is an important metric that should be discussed, especially considering the additional computational overhead introduced by the approach.
5. I am also curious about the sensitivity of the threshold selection strategy. It doesn’t seem highly sensitive, but how does it perform over a broader parameter range or with different thresholding strategies? This could be a point worth discussing in the paper.

If the authors can adequately respond to these concerns, I would consider increasing my score.

---

> ### Author Response · Authors · 2024-11-24
> **Response to Reviewer LuhJ (1/4)**
>
> Thank you for taking the time to review our paper and offering insightful feedback. In response to your concerns, we would like to provide the following explanations.
>
> 1. **Q1:** The novelty of this paper should be emphasized more clearly. From the motivation perspective, dividing confident and non-confident samples and applying progressive training is a fairly conventional approach. Similar ideas have been extensively used in domain adaptation (DA) problems, and several papers in TTA focus on pseudo-labeling.
>
>    **A1:** The key contribution of our method is the first introduction of uncertain supervision into the online TTA (OTTA) paradigm, and using candidate label sets is one effective approach within this broader framework. Online learning, characterized by the model encountering each sample and its associated supervision only once, makes any resulting impact on the model permanent and irreversible. During the adaptation process, it is difficult for the model to generate definite and correct supervision for all samples. However, previous online TTA methods tend to rashly adopt definite supervision to guide the model adaptation and overlook the irreversible detrimental effects of the introduction of false supervision. Our article seeks to address this urgent and critical issue in the online TTA paradigm by introducing uncertain supervision. By employing candidate label sets as one approach, the model gradually enhances its performance by leveraging uncertain supervision while avoiding the interference of incorrect supervision during the online process.
>
>    Our work is the first to employ a theoretically guaranteed candidate labeling approach in the online TTA setting. Additionally, the domain adaptation setting differs from online TTA. While the two share some conceptual similarities in method design, they address fundamentally different problems. Furthermore, the pseudo-labels used in online TTA contrast substantially with the candidate labels proposed in this work, particularly in terms of label certainty.

---

> ### Author Response · Authors · 2024-11-24
> **Response to Reviewer LuhJ (2/4)**
>
> 2. **Q2:** Based on the motivation of the paper, if the so-called more effective supervised information can be quantified? If pseudo-label error terms or confidence levels could be incorporated, it would help reveal how the label-generation process impacts generalization performance, thereby offering more practical insights. Additionally, how is the divergence term in the bound reduced in this paper? How does it influence pseudo-labeling and progressive adaptation?
>
>    **A2:** We further provide a new theorem that assesses the effectiveness of pseudo-labels by quantifying them through pseudo-label error terms for TTA, and we have updated theorem 2 and proof in the revised version. Assume that during test-time adaptation, the predictive model streamingly receives $R$ mini-batch data from the target domain $T$, accumulating a dataset $\mathcal{D}^R_T$ over $R$ mini-batches, with a total sample size of $N^R$. For a target domain sample $\boldsymbol{x}$, let its Bayes class-probability distribution be denoted as $\boldsymbol{p}=\left[P\left(y_{1} \mid \boldsymbol{x}\right), P\left(y_{2} \mid \boldsymbol{x}\right), \ldots, P\left(y_{c} \mid \boldsymbol{x}\right)\right]$, and its supervision provided by the algorithm be denoted as $\boldsymbol{q}$ (here, it refers to the label distribution). We have the following theorem:
>
>    Theorem 2. Suppose the loss function $\ell$ is bounded by $M$, i.e., $M=\sup_{\boldsymbol{x} \in \mathcal{X}, f \in \mathcal{F}, y_{j} \in \mathcal{Y}} \ell(f(\boldsymbol{x}), y)$. Fix a hypothesis class $\mathcal{F}$ of predictors $f: \mathcal{X} \mapsto \mathbb{R}^{c}$, with induced class $\mathcal{H} \subset[0,1]^{\mathcal{X}}$ of functions $h(\boldsymbol{x})=\ell\left(f\left(\boldsymbol{x} _ {i}\right), \boldsymbol{q}\right)$. Suppose $\mathcal{H}$ has uniform covering number $\mathcal{N}_{\text {inf }}$. Then for any $\delta \in(0,1)$, with probability at least $1-\delta$,
>
>    $$
>    R(f)-\widehat{R}(f) \leq M \sqrt{c} \cdot\left(\mathbb{E}\left[\|\boldsymbol{q}-\boldsymbol{p}\| _ {2}\right]\right)  +\mathcal{O}\left(\sqrt{\mathbb{V}(f) \cdot \frac{\log \frac{\mathcal{M} _ {N^R}}{\delta}}{N^R}}+\frac{\log \frac{\mathcal{M} _ {N^R}}{\delta}}{N^R}\right)
>    $$
>
>    where $\mathcal{M} _ {N^R}=\mathcal{N} _ {\inf}\left(\frac{1}{N^R}, \mathcal{H}, 2 N^R\right)$, and $\mathbb{V}(f)$ is the empirical variance of the loss values.
>
>    Theorem 2 demonstrates that as the target domain samples' label distribution $\boldsymbol{q}$ provided by the algorithm becomes closer to the Bayes class-probability distribution $\boldsymbol{p}$, the gap between the empirical risk and the expected risk on the accumulated dataset $\mathcal{D}^R_T$ will decrease. The effectiveness of the supervision can be quantified by the degree of closeness between its corresponding label distribution and the Bayes class-probability distribution and the pseudo-label error terms is $\mathbb{E}\left[\|\boldsymbol{q}-\boldsymbol{p}\|_{2}\right]$. Our algorithm provides one-hot pseudo-labels when it is certain about the samples, and a candidate pseudo-label set when it is uncertain. These actions can make the corresponding pseudo-label's label distribution closer to the Bayes class-probability distribution, thereby making the empirical risk more closely aligned with the expected risk, and thus better guiding the model towards adaptation to the target domain.
>
>    Regarding the bound provided in Theorem 1, for a specific classifier $h$, $d_{h, \mathcal{H}}(S, T)$ is a constant that quantifies the discrepancy between $S$ and $T$, under the assumption that the hypothesis space $\mathcal{H}$, the source domain and the target domain remain fixed. Pseudo-label-based TTA methods reduce this generalization error bound by gradually training the model on streaming data, enabling it to encounter more samples with effective supervision. This effect is reflected in the role of $m_r$ within the bound.
>
>    A larger $d_{h, \mathcal{H}}(S, T)$ results in a looser generalization error bound for the error minimizer, making it more challenging to guarantee effective adaptation. Our algorithm addresses this by incorporating uncertain supervision and a temporary storage mechanism to ensure the reliability and sufficiency of the supervision. Additionally, we initialize the threshold conservatively to account for the significant distribution gap at the start of adaptation and gradually decrease it as the model aligns more closely with the target domain.

---

> ### Author Response · Authors · 2024-11-24
> **Response to Reviewer LuhJ (3/4)**
>
> 3. **Q3:** Regarding the experimental setup, the datasets used in this paper differ from those employed in previous methods. The rationale for these choices should be explained in detail. Furthermore, for certain methods with the same settings like PROGRAM, why do the results differ from the original paper when using the same benchmark and backbone? Could it be due to different settings or other reasons?
>
>    **A3:** The datasets used in this study are also widely employed by many recent online TTA methods, such as TSD [1], TAST [2], and PROGRAM [3]. These methods similarly evaluate the effectiveness of TTA approaches using both domain generalization datasets and image corruption datasets. We believe the six datasets, encompassing different numbers of classes and diverse types of distribution shifts, are representative. For all baselines, we re-evaluated them using their official implementations provided by the authors. For PROGRAM, since its code is not yet publicly available, we implemented it ourselves. If needed, we are willing to make our implementation code publicly available.
>
>
> 4. **Q4:** The paper lacks ablation studies to evaluate the effectiveness of each module. Additionally, since the proposed method is an online model, time efficiency is an important metric that should be discussed, especially considering the additional computational overhead introduced by the approach.
>
>    **A4:** We additionally conducted ablation studies using two simplified variants of our framework: PASLE-NB and PASLE-NR. In PASLE-NB, the buffer is removed from the framework, while in PASLE-NR, the strategy of threshold reduction is excluded. For this study, we utilized the OfficeHome dataset and employed ResNet-18 as the backbone. The results are shown in the following tables, highlighting that the buffer mechanism and threshold reduction strategy, as pluggable modules in our framework, further improve its performance.
>
>    **Table 1:** Classification accuracy of PASLE and its variants (PASLE-NB and PASLE-NR) on the OfficeHome dataset.
>
>    |          | A          | C          | P          | R          |
>    | -------- | ---------- | ---------- | ---------- | ---------- |
>    | PASLE    | 57.25±0.75 | 51.30±0.41 | 73.31±1.04 | 74.10±0.20 |
>    | PASLE-NB | 56.98±0.82 | 51.14±0.44 | 73.14±0.87 | 73.00±0.25 |
>    | PASLE-NR | 57.02±0.76 | 51.11±0.39 | 73.09±1.21 | 72.98±0.33 |
>
>    Regarding computational overhead, the primary additional cost of our method lies in the forward propagation of samples stored in the buffer. However, the number of samples in the buffer does not always reach its maximum capacity, and we selectively perform backpropagation for these samples, saving a portion of the computational cost. Furthermore, other modules, such as pseudo-label generation and threshold reduction, incur minimal computational overhead. To evaluate the computational cost, experiments were carried out on the clipart domain of the DomainNet dataset, using ResNet-18 as the backbone with a batch size of 128 on an NVIDIA TITAN Xp GPU. The reported runtime excludes data loading time, ensuring fairness by using torch.cuda.synchronize() to accurately measure the computational overhead. The results are shown in the following table. It can be observed that our method does not incur significantly more computational overhead compared to other state-of-the-art methods.
>
>    **Table 2:** Running time of different methods on the clipart domain of DomainNet dataset.
>
>    | Baseline | Time (s) |
>    | -------- | -------- |
>    | ERM      | 19.54    |
>    | BN       | 21.03    |
>    | TENT     | 57.23    |
>    | PL       | 77.94    |
>    | SHOT-IM  | 77.02    |
>    | T3A      | 46.57    |
>    | TAST     | 86.89    |
>    | TAST-BN  | 128.46   |
>    | TSD      | 105.55   |
>    | PROGRAM  | 113.58   |
>    | DeYO     | 92.49    |
>    | PASLE    | 99.28    |

---

> ### Author Response · Authors · 2024-11-24
> **Response to Reviewer LuhJ (4/4)**
>
> 5. **Q5:** I am also curious about the sensitivity of the threshold selection strategy. It doesn’t seem highly sensitive, but how does it perform over a broader parameter range or with different thresholding strategies?
>
>    **A5:** We conducted a parameter sensitivity analysis experiment on the CIFAR-10-C dataset under shot noise with a broader range of hyperparameters. The value of $\tau_{start}$ was selected from a wider range, specifically between 0.2 and 0.9. The threshold gap represented as $|\tau_{start} - \tau_{end}|$, was fixed at 0.1. For testing purposes, $\tau_{des}$ was set to $\frac{\tau_\text{start} - \tau_\text{end}}{R}$. The results are summarized in the table below.
>
>    **Table 3:** Classification accuracy of PASLE under broader parameter range.
>
>    | $\tau_{start}$ | $\tau_{end}$ | Acc   |
>    | -------------- | ------------ | ----- |
>    | 0.9            | 0.8          | 77.90 |
>    | 0.8            | 0.7          | 77.97 |
>    | 0.7            | 0.6          | 77.96 |
>    | 0.6            | 0.5          | 77.99 |
>    | 0.5            | 0.4          | 77.89 |
>    | 0.4            | 0.3          | 77.84 |
>    | 0.3            | 0.2          | 77.84 |
>    | 0.2            | 0.1          | 77.79 |
>
>    The results indicate that the algorithm achieves optimal performance when $\tau$ is within the range of 0.5 to 0.8. Within a reasonable range of $\tau$, the algorithm also delivers comparable results. However, when $\tau$ is set too low (e.g., within the range of 0.1 to 0.2), many samples with incorrect supervision are introduced, leading to a decline in performance.
>
>    Additionally, we experimented with scheduling the threshold using a cosine function. The decay strategy was defined as $\tau(r) = \tau_{end} + (\tau_{start} - \tau_{end}) \cdot \frac{1 + \cos(\pi \cdot r / R)}{2}$. The results in the table below show that the cosine scheduling approach generally leads to slightly lower performance compared to the linear threshold decay.
>
>    **Table 4:** Classification accuracy of PASLE under different thresholding strategies.
>
>    | $\tau_{start}$ | $\tau_{end}$ | Linear Acc | Cosine Acc |
>    | -------------- | ------------ | ---------- | ---------- |
>    | 0.9            | 0.8          | 77.90      | 77.95      |
>    | 0.8            | 0.7          | 77.97      | 77.87      |
>    | 0.7            | 0.6          | 77.96      | 77.86      |
>    | 0.6            | 0.5          | 77.99      | 77.96      |
>    | 0.5            | 0.4          | 77.89      | 77.85      |
>
> **Refs.**
>
> [1] Feature Alignment and Uniformity for Test Time Adaptation. CVPR 2023
>
> [2] Test-Time Adaptation via Self-Training with Nearest Neighbor Information. ICLR 2023
>
> [3] PROGRAM: PROtotype GRAph Model based Pseudo-Label Learning for Test-Time Adaptation. ICLR 2024

---

> > ### Comment · Reviewer_LuhJ · 2024-11-25
> > **Question restatement**
> >
> > Thank you for the patient response of the authors. It seems I may not have expressed myself clearly, as the replies did not fully address my concerns. My main focus—whether in terms of novelty, ablation studies, or sensitivity analysis—centers on the introduction of uncertainty, which I see as the core of this paper. Regarding novelty, the authors highlight the differences between online TTA, TTA, and DA. However, I am curious about the distinctions of uncertainty-based methods within these scenarios. My focus is not on the differences between the scenarios themselves, but rather on the connections between these uncertainty-based methods and what makes the proposed approach unique. Similarly, for the ablation studies, I am primarily interested in understanding the impact of uncertainty modeling. However, the response is not sufficiently clear about this aspect.
> > Additionally, regarding the analysis of time complexity, while the authors provided experimental results, I am more curious about the underlying reasons behind these results. For instance, I initially assumed that the proposed method would incur relatively high theoretical overhead. However, the experimental results show only about a 20-second difference compared to commonly used methods like SHOT and, in fact, even outperform some methods in terms of speed. Why is this the case? This is what I would like to understand better.

---

> > > ### Author Response · Authors · 2024-11-30
> > >
> > > Dear Reviewer LuhJ,
> > >
> > > Thank you once again for your valuable feedback. If you have any questions or require further clarification, please do not hesitate to reach out. We will respond promptly to address your concerns. We deeply appreciate your time and effort in reviewing our work. Your feedback is invaluable to us, and we eagerly await your insights.
> > >
> > > Best regards,
> > >
> > > Authors

---

> > > > ### Comment · Reviewer_LuhJ · 2024-12-02
> > > >
> > > > Thanks for the author's response to these concerns, I will increase my score.

---

> ### Author Response · Authors · 2024-11-26
> **Response to Questions**
>
> We sincerely appreciate your time and effort in reviewing our manuscript, and hope that the following responses will be able to address your concern.
>
> 1. **About the distinctions of uncertainty-based methods within these scenarios (online TTA, TTA, DA)**
>
> Our uncertainty-based method to deal with online TTA is very different from those within DA or TTA in uncertainty modeling. Our proposed method models the uncertainty at the label level through candidate pseudo-label sets, while the previous methods within DA or TTA build up the uncertainty at the model or sample level. For example, at the model level, [1] leverages Monte Carlo (MC) dropout, and [2] applies deep ensembles. At the sample level, [3] generates sample weight through the uncertainty.
>
> Compared to the previous methods, our proposed method, where the uncertainty is directly manifested through the cardinality of the candidate pseudo-label set, has no limitation on the pre-trained model, a higher sample utilization rate, and a theoretical guarantee, which is more suitable to deal with online TTA.
>
> Besides, as refered by Q1, pseudo-labeling in TTA such as [4] divides confident and non-confident samples to perform entropy minimization on confident samples with correct pseudo-labels rather than explicitly model uncertainty on non-confident samples.
>
> **Refs.**
>
> [1] Test-time Adaptation for Machine Translation Evaluation by Uncertainty Minimization. ACL 2023
>
> [2] Hypothesis disparity regularized mutual information maximization. AAAI 2021
>
> [3] Uncertainty-guided Source-free Domain Adaptation. ECCV 2022
>
> [4] Feature Alignment and Uniformity for Test Time Adaptation. CVPR 2023
>
> 2. **About the underlying reasons behind the experimental results of time complexity**
>
> Here, we analyze the underlying reasons behind the experimental results of time complexity.
>
> For methods faster than ours, the reasons can be summarized as follows:
>
> 1. Not updating model parameters (ours vs ERM, T3A).
> 2. Updating only a subset of model parameters (ours vs BN, TENT, TAST).
> 3. Processing fewer samples compared to our method (ours vs PL, SHOT-IM, DeYO).
>
> For methods slower than ours, the primary reason lies in the additional overhead caused by kNN computations (ours vs TSD, TAST-BN, PROGRAM).
>
> In fact, our method has the similar time complexity with the baseline PL in pseudo-label generation. The reason why it is a bit slower than PL is that our sample utilization rate is high, and as a result, better performance has been achieved.

---

### Official Review · Reviewer_mB9B · 2024-10-30

**Soundness:** 3
**Presentation:** 3
**Contribution:** 2
**Rating:** 6
**Confidence:** 4

**Summary:**

The problem studied in this paper is the conventional test-time adaptation. When assigning pseudo-labels to test samples, the paper assigns one label to samples with high confidence, while assigning a candidate set of labels to less confident samples. It uses a buffer to store samples that could not be labeled, allowing the model to attempt labeling them in subsequent batches. Finally, the model is updated by using cross-entropy with the one-hot encoded pseudo-labels. The effectiveness of the method is validated across multiple datasets.

**Strengths:**

1.	The writing of this paper is clear, and both the problem definition and the method description are well articulated.
2.	The motivation behind the proposed label enhancement method is reasonable, and there is substantial theoretical analysis provided.
3.	The experiments in the paper are relatively thorough, demonstrating the superiority of the proposed method.

**Weaknesses:**

1.	Despite the relatively comprehensive theoretical analysis, the design of the method in this paper is overly simplistic. Similar approaches using candidate pseudo-label sets have long existed in the field of semi-supervised learning.
2.	The maintenance of this buffer seems somewhat unfair. If a sample’s label remains undecided for an extended period, it will be repeatedly seen by the model in subsequent iterations. Although the buffer size imposes some constraints, the repeated processing of test samples could still introduce bias. Additionally, maintaining a buffer incurs significant overhead. If the buffer becomes too large, the number of samples to be predicted in each batch will be dictated more by the buffer size than by the batch size itself.

**Questions:**

1.	Since there are many approaches for creating pseudo-label candidate sets, has the paper compared its method with other approaches for selecting pseudo-label candidates? Does this method have any unique advantages specifically for the test-time adaptation (TTA) task? Or is it also applicable to semi-supervised or unsupervised tasks?
2.	What is the buffer size used in the experiments? Was there any ablation study conducted on the buffer size? If the buffer were removed, would this method still be effective?
3.	In the experiment section, why do ERM and T3A perform so poorly on CIFAR10-C and CIFAR100-C? In the original papers and subsequent TTA studies, their performance was not as weak.

---

> ### Author Response · Authors · 2024-11-24
> **Response to Reviewer (1/3)**
>
> Thank you for taking the time to thoroughly review our paper and provide valuable feedback. In response to your concerns, we would like to provide the following explanations.
>
> 1. **Q1:** Despite the relatively comprehensive theoretical analysis, the design of the method in this paper is overly simplistic. Similar approaches using candidate pseudo-label sets have long existed in the field of semi-supervised learning.
>
>    **A1:** In the field of online TTA (OTTA), designing methods that are both effective and simple is particularly important, as models require real-time adaptation. Excessive computational overhead from overly complex designs is unacceptable in scenarios like autonomous driving.
>
>    The key contribution of our method is the first introduction of uncertain supervision into the online TTA paradigm while using candidate label sets is one effective approach within this broader framework. Online learning, characterized by the model encountering each sample and its associated supervision only once, makes any resulting impact on the model permanent and irreversible. During the adaptation process, it is difficult for the model to generate definite and correct supervision for all samples. However, previous online TTA methods tend to rashly adopt definite supervision to guide the model adaptation and overlook the irreversible detrimental effects of the introduction of false supervision. Our article seeks to address this urgent and critical issue in the online TTA paradigm by introducing uncertain supervision. By employing candidate label sets as one approach, the model gradually enhances its performance by leveraging uncertain supervision while avoiding the interference of incorrect supervision during the online process.
>
>    Moreover, the semi-supervised learning setting differs from online TTA in that it includes a portion of labeled samples, which are absent in online TTA. This difference results in variations in the candidate label generation process. Furthermore, the incorporation of uncertainty and candidate labels in our method is not heuristic but is grounded in and guided by solid theoretical guarantees specifically derived for the online TTA setting.
>
> 2. **Q2:** The maintenance of this buffer seems somewhat unfair. Although the buffer size imposes some constraints, the repeated processing of test samples could still introduce bias.
>
>    **A2:** The buffer is a commonly used and fair technique in OTTA, as demonstrated by the memory bank in TSD [1] and AdaNPC [2], as well as the support set in TAST [3] and T3A [4]. Essentially, all of these are variations of a buffer. Besides, the buffer is a modular component within our framework, designed to temporarily store a portion of samples for potential future use, rather than being a non-decouplable component. As shown in Table 1 of the ablation study provided in A3, our framework maintains a notable performance advantage over other methods, even without incorporating a buffer.
>
>    Since samples that cannot provide effective supervision are not used to update model parameters and only have a minimal effect on the statistics of the BN layers, they are unlikely to introduce bias. Besides, we select the samples with the top-$K$ largest margins to store in the buffer, as their effective supervision is likely to emerge earlier compared to other samples. This approach also ensures the dynamic flow of samples within the buffer.

---

> ### Author Response · Authors · 2024-11-24
> **Response to Reviewer mB9B (2/3)**
>
> 3. **Q3:** What is the buffer size used in the experiments? Was there any ablation study conducted on the buffer size? If the buffer were removed, would this method still be effective? Additionally, maintaining a buffer incurs significant overhead.
>
>    **A3:** In the paper, we mentioned: “The buffer’s maximum capacity $K$ is restricted to a quarter of the target domain batch size in practice” (Line 232, Page 6), and “The batch size for the online target domain data is set to 128” (Line 411, Page 8). Therefore, the buffer’s maximum capacity $K$ in our experiments is 32. Below, we present the results of experiments conducted on the CIFAR-10-C dataset with buffer sizes of 16 and without a buffer in our framework. For this study, a subclass was randomly selected from four different types of corruption (Noise, Blur, Weather, Digital). The results are shown in the table below:
>
>    **Table 1:** Classification accuracy of PASLE with different buffer capacity on CIFAR-10-C dataset.
>
>    |            | Shot noise | Zoom blur | Fog   | Pixelation |
>    | ---------- | ---------- | --------- | ----- | ---------- |
>    | PASLE K=32 | 78.03      | 80.81     | 72.31 | 81.16      |
>    | PASLE K=16 | 77.95      | 80.72     | 72.24 | 81.10      |
>    | PASLE K=0  | 77.88      | 80.61     | 72.10 | 80.98      |
>    | SHOT-IM    | 77.20      | 79.90     | 71.35 | 80.54      |
>
>    The results show that even without a buffer, our algorithm still significantly outperforms SHOT-IM (the second-best method in Table 2 of our paper).
>
>    To evaluate the computational cost introduced by the buffer, we conducted tests on all baselines, including PASLE and a vanilla variant, PASLE-NB, which excludes the buffer from our framework. The experiments were carried out on the clipart domain of the DomainNet dataset, using ResNet-18 as the backbone with a batch size of 128 on an NVIDIA TITAN Xp GPU. The reported runtime excludes data loading time, ensuring fairness by using the torch.cuda.synchronize() to accurately measure the computational overhead. The buffer’s maximum capacity $K$ is set to one-quarter of the batch size. The results, presented in the table below, indicate that the inclusion of the buffer incurs only manageable additional overhead.
>
>    **Table 2:** Running time of different methods on the clipart domain of DomainNet dataset.
>
>    | Baseline | Time (s) |
>    | -------- | -------- |
>    | ERM      | 19.54    |
>    | BN       | 21.03    |
>    | TENT     | 57.23    |
>    | PL       | 77.94    |
>    | SHOT-IM  | 77.02    |
>    | T3A      | 46.57    |
>    | TAST     | 86.89    |
>    | TAST-BN  | 128.46   |
>    | TSD      | 105.55   |
>    | PROGRAM  | 113.58   |
>    | DeYO     | 92.49    |
>    | PASLE    | 99.28    |
>    | PASLE-NB | 79.59    |

---

> ### Author Response · Authors · 2024-11-24
> **Response to Reviewer mB9B (3/3)**
>
> 4. **Q4:** Since there are many approaches for creating pseudo-label candidate sets, has the paper compared its method with other approaches for selecting pseudo-label candidates? Does this method have any unique advantages specifically for the test-time adaptation (TTA) task? Or is it also applicable to semi-supervised or unsupervised tasks?
>
>    **A4:** We further explored two approaches for generating candidate labels. The first is a threshold-based approach (PASLE-TB), where a threshold is set, and all classes with prediction probabilities exceeding this threshold are selected as candidate labels. This method generates both one-hot pseudo-labels and candidate pseudo-label sets. The second approach is top-K based (PASLE-KB), where prediction probabilities are sorted in descending order, and the top-K classes are chosen as candidate labels. Unlike the first method, this approach only produces candidate pseudo-label sets. The threshold and K are dynamically adjusted during the adaptation process. Experiments were conducted on the OfficeHome dataset using ResNet-18 as the backbone, and the results are presented in the table below.
>
>    **Table 3:** Classification accuracy of PASLE with different candidate label selection strategies on OfficeHome dataset.
>
>    |          | A     | C     | P     | R     |
>    | -------- | ----- | ----- | ----- | ----- |
>    | PASLE    | 57.25 | 51.30 | 73.31 | 74.10 |
>    | PASLE-TB | 57.07 | 51.13 | 73.11 | 73.92 |
>    | PASLE-KB | 56.31 | 50.69 | 72.82 | 73.56 |
>
>    It can be observed that PASLE-TB achieves performance comparable to PASLE, while PASLE-KB, which lacks sample selection and directly uses the top-K predicted classes of all samples as candidate labels, performs significantly worse than PASLE.
>
>    The central innovation of our method lies in introducing uncertain supervision to the online TTA paradigm, with candidate label sets serving as a practical and impactful implementation within this framework. Moreover, our method can be adapted for semi-supervised learning with appropriate modifications. For unlabeled samples, candidate labels can be generated using the approach outlined in Proposition 1, with thresholds dynamically adjusted based on the training process. In addition, the thresholds can be adjusted separately for each class based on its level of difficulty, tailored to the characteristics of semi-supervised learning.
>
> 5. **Q5:** In the experiment section, why do ERM and T3A perform so poorly on CIFAR10-C and CIFAR100-C? In the original papers and subsequent TTA studies, their performance was not as weak.
>
>    **A5:** We adopted the same experimental setup as recent baselines and ensured that all methods were re-evaluated under this setup. As the latest baselines (TSD [1], TAST [3], PROGRAM [5]) have not released their code for CIFAR-C dataset, there might be minor setting differences in the training process of the pre-trained source models, such as learning rate decay settings. Since ERM and T3A both directly rely on the source model without updating its parameters during testing, their extracted features and predictions are highly susceptible to distribution shifts, resulting in poor performance. In contrast, even minimal adjustments to the model’s BN layers significantly improve performance on the target domain, as demonstrated by the baseline BN results.
>
> **Refs.**
>
> [1] Feature Alignment and Uniformity for Test Time Adaptation. CVPR 2023
>
> [2] AdaNPC: Exploring Non-Parametric Classifier for Test-Time Adaptation. ICML 2023
>
> [3] Test-Time Adaptation via Self-Training with Nearest Neighbor Information. ICLR 2023
>
> [4] Test-Time Classifier Adjustment Module for Model-Agnostic Domain Generalization. NeurIPS 2021
>
> [5] PROGRAM: PROtotype GRAph Model based Pseudo-Label Learning for Test-Time Adaptation. ICLR 2024

---

> ### Author Response · Authors · 2024-11-28
>
> Dear reviewer mB9B,
>
> Thank you very much for your constructive comments on our work. We've made every effort to address the concerns raised. As the discussion period is nearing its conclusion, could we kindly inquire if you have any remaining questions or concerns? Thanks for your efforts in reviewing our work, and we sincerely look forward to your reply.
>
> Best regards,
>
> Authors

---

### Official Review · Reviewer_z3cq · 2024-11-03

**Soundness:** 3
**Presentation:** 4
**Contribution:** 3
**Rating:** 8
**Confidence:** 5

**Summary:**

This paper studies Test-time adaptation (TTA), which aims to adapt a pre-trained model to the target domain using only unlabeled test samples. The authors proposed a new TTA framework, which assigns candidate pseudo-label sets to uncertain ones via selective label enhancement. The model is progressively trained on certain and uncertain pseudo-labeled data while dynamically refining uncertain pseudo-labels, leveraging increasing target adaptation monitored throughout training. Experiments on various benchmark datasets validate the effectiveness of the proposed approach.

**Strengths:**

- Instead of assigning definite pseudo-labels to test samples, candidate pseudo-label sets are assigned to uncertain ones via selective label enhancement.
- The proposed method partitions test samples into confident and uncertain subsets based on the model’s predictive confidence scores, with confident samples receiving one-hot pseudo-labels, and uncertain samples being assigned candidate pseudo-label sets
- The theory establishes a generalization bound for TTA that by incorporating a greater number of target domain samples with effective supervision, a tighter generalization bound can be achieved.

**Weaknesses:**

- In the proposed method, the authors need to provide more details about the reduced threshold to improve the reliability of pseudo labels.
- Why use image corruption datasets to validate the effectiveness of the proposed method? 15 types of common image corruptions should be shown clearly.
- This paper uses a vanilla variant that all samples annotated with candidate pseudo-labels sets excluded from model updates to demonstrate the effectiveness of the candidate pseudo-labels sets of the proposed method. More detests could be added in this ablation experiments.

**Questions:**

- In the proposed method, why the authors reduced to improve the reduced threshold could improve the reliability of pseudo labels.
- In the experiments, why did the authors only adopt online test-time adaptation approaches as the baselines?

---

> ### Author Response · Authors · 2024-11-24
> **Response to Reviewer z3cq**
>
> Thank you for dedicating your time to reviewing our paper and offering insightful feedback. In response to your concerns, we would like to provide the following explanations.
>
> 1. **Q1:** Why the authors reduced to improve the reduced threshold could improve the reliability of pseudo labels? The authors need to provide more details about the reduced threshold to improve the reliability of pseudo labels.
>
>    **A1:** The threshold characterizes the distance between the current model and the Bayesian optimal classifier for the target domain through the difference in the probability of each class of a sample. As the model adapts under effective supervision, this gap gradually decreases, leading to a reduction in the probability differences for each class. If the threshold does not decrease accordingly, candidate labels that should have been excluded might instead be selected. Even if the correct label is included in the candidate set, this could result in redundant candidate labels. Moreover, samples that could otherwise provide effective supervision might become unusable due to the candidate set encompassing all possible classes. This would weaken the effectiveness of model adaptation. Therefore, the gradual reduction of the threshold is essential.
>
> 2. **Q2:** Why use image corruption datasets to validate the effectiveness of the proposed method? 15 types of common image corruptions should be shown clearly.
>
>    **A2:** In our study, we utilize two image corruption datasets, CIFAR-10C and CIFAR-100C, which are generated by applying corruptions to the original clean CIFAR-10 and CIFAR-100 datasets. These datasets have been widely adopted by many online TTA methods to evaluate algorithm performance under distribution shifts, including recent approaches like PROGRAM [1] and DeYO [2]. Corruptions consist of 15 different types categorized into four groups:
>
>    - Noise: Gaussian noise, Shot noise, Impulse noise
>
>    - Blur: Defocus blur, Glass blur, Motion blur, Zoom blur
>
>    - Weather: Snow, Frost, Fog, Brightness
>
>    - Digital: Contrast, Elastic transformation, Pixelation, and JPEG compression.
>
>    Each type of corruption has five severity levels, where a larger severity level indicates a more pronounced distribution shift.
>
> 3. **Q3:** More detests could be added in the ablation experiments about PASLE-NC.
>
>    **A3:** We conducted additional ablation studies on two domain adaptation datasets, PACS and DomainNet, utilizing ResNet-18 as the backbone. The results are presented in the table below:
>
>    **Table 1:** Classification accuracy of PASLE and PASLE-NC on PACS dataset.
>
>    |          | A          | C          | P          | S          |
>    | -------- | ---------- | ---------- | ---------- | ---------- |
>    | PASLE    | 88.19±1.42 | 87.09±0.24 | 96.83±0.48 | 80.51±1.28 |
>    | PASLE-NC | 86.82±1.52 | 85.98±0.57 | 96.23±0.44 | 79.56±1.50 |
>
>    **Table 2:** Classification accuracy of PASLE and PASLE-NC on DomainNet dataset.
>
>    |          | clipart    | infograph  | painting   | quickdraw  | real       | sketch     |
>    | -------- | ---------- | ---------- | ---------- | ---------- | ---------- | ---------- |
>    | PASLE    | 51.76±0.39 | 14.98±0.28 | 43.06±0.15 | 13.67±0.26 | 52.69±0.18 | 45.15±0.27 |
>    | PASLE-NC | 50.82±0.23 | 13.18±0.34 | 42.29±0.14 | 13.28±0.39 | 51.99±0.36 | 44.21±0.31 |
>
> 4. **Q4:** In the experiments, why did the authors only adopt online test-time adaptation approaches as the baselines?
>
>    **A4:** Our study focuses on online test-time adaptation, and as such, we primarily compare our approach against other online test-time adaptation methods. In contrast, source-free domain adaptation methods typically require access to the entire target domain dataset and perform multiple rounds of adaptation. The online TTA paradigm, where the model processes one batch of data at a time and requires immediate updates, often demands more specialized method designs to address its unique challenges.
>
> **Refs.**
>
> [1] PROGRAM: PROtotype GRAph Model based Pseudo-Label Learning for Test-Time Adaptation. ICLR 2024
>
> [2] Entropy is not Enough for Test-Time Adaptation: From the Perspective of Disentangled Factors. ICLR 2024

---

> > ### Comment · Reviewer_z3cq · 2024-11-25
> >
> > Thanks for your feedback. My concerns have been addressed, and I keep the scores.

---

### Official Review · Reviewer_oudU · 2024-11-04

**Soundness:** 3
**Presentation:** 2
**Contribution:** 2
**Rating:** 6
**Confidence:** 3

**Summary:**

The paper introduces a new pseudo-learning algorithm that combines one-hot label learning and candidate label learning approaches. Each learning paradigm is conducted on its respective sample set, referred to as the certain set for one-hot label learning and the uncertain set for candidate learning. The key distinction from other pseudo-label learning papers is that the authors propose a theoretical guarantee to ensure that the selected labels in the pseudo-set will correspond to the ground truth if certain conditions are met (Proposition 1). In the initial learning stages, the model focuses more on candidate set learning and gradually shifts toward minimizing the one-hot label loss as it updates more on target samples. The authors provide a theory indicating that the generalization bound becomes tighter as more target samples are incorporated. Experimental results demonstrate the algorithm's performance compared to other TTA learning methods.

**Strengths:**

1. The proposed method is supported with a theory guarantee.
2. The experiment is diverse datasets, which confirm the effectiveness of the proposed methods.

**Weaknesses:**

The reviewer's main concern is the novelty of the proposed approach: adapting pseudo-learning and candidate learning is already popular in TTA and domain adaptation, as the authors discussed in Section 2. The main novelty here comes from Proposition 1, where the authors propose to ensure the correctness of pseudo labels under specific assumptions. The condition is that the learned weight and the optimal one need to be close enough (the closeness is measured by the difference in the probability of each class in the input samples, and $\tau(r)$ is the threshold). The selected pseudo labels are considered true when this condition is met. However, how can we ensure that this condition is always satisfied? If the reviewers understand correctly, this condition is based on the threshold $\tau(r)$, which is initialized to 1 and then gradually reduced to a specific value. When $\tau(r)$ is smaller than 1, how can we ensure that the distance between the learned models and the optimal one is smaller than this threshold?

**Questions:**

Please refer to the Weakness.

---

> ### Author Response · Authors · 2024-11-24
> **Response to Reviewer oudU**
>
> Thank you for taking the time to review our paper and providing valuable feedback. In response to your concerns, we would like to provide the following explanations.
>
> 1. **Q1:** The reviewer's main concern is the novelty of the proposed approach: adapting pseudo-learning and candidate learning is already popular in TTA and domain adaptation.
>
>    **A1:** To the best of our knowledge, our work is the first one to apply candidate pseudo-label sets to online test-time adaptation (OTTA), which considers uncertainty ignored by previous OTTA pseudo-labeling work and provides a theoretical guarantee for the generation of candidate pseudo-label sets rather than rely on a heuristic approach. Currently, there is only one study employing candidate labels in the context of unsupervised domain adaptation [1], which was published after our submission. Furthermore, the unsupervised domain adaptation setting, which permits access to labeled source domain data, is fundamentally different from OTTA, which has no access to labeled source domain data.
>
>    Besides, the key contribution of our method is the first introduction of uncertain supervision into the online TTA paradigm, while using candidate label sets is one effective approach within this broader framework. Online learning involves the model encountering each sample and its associated supervision only once, making any impact permanent and irreversible. To address the risks of false supervision in online TTA, our method introduces uncertain supervision, using candidate label sets to enhance performance while mitigating the effects of incorrect supervision.
>
> 2. **Q2:** The selected pseudo labels are considered true when this condition is met. However, how can we ensure that this condition is always satisfied?
>
>    **A2:** The threshold used in our algorithm is an estimation of the theoretical threshold $\tau$. To ensure the robustness of this estimation, we set a relatively conservative $\tau$ at the beginning of adaptation to include correct labels within the uncertain supervision. To ensure the effectiveness of the estimation, the threshold is dynamically reduced as the model aligns more closely with the target domain, allowing the generated supervision to more efficiently assist in adaptation. Consequently, the threshold employed by our algorithm provides a reasonable and robust approximation of the theoretical $\tau$, contributing to the performance improvements observed in our experiments. Furthermore, sensitivity analysis of the threshold demonstrates the strong adaptability of our framework to various threshold configurations.
>
> **Ref.**
>
> [1] Improving Unsupervised Domain Adaptation: A Pseudo-Candidate Set Approach. ECCV 2024

---

> ### Author Response · Authors · 2024-11-28
>
> Dear reviewer oudU,
>
> Thank you very much for your constructive comments on our work. We've made every effort to address the concerns raised. As the discussion period is nearing its conclusion, could we kindly inquire if you have any remaining questions or concerns? Thanks for your efforts in reviewing our work, and we sincerely look forward to your reply.
>
> Best regards,
>
> Authors

---

> > ### Comment · Reviewer_oudU · 2024-12-02
> >
> > Dear Authors, sorry for the late reply,
> >
> > I have read your response to my concerns, and your discussion with another reviewer. I am partially satisfied with your response to my second concern because the term robustness you indicate is just a heuristic approach, which may not always work. However, you have addressed a lot of other problems, where I think the current paper quality has reached the acceptance bar of ICLR. So I decide to raise my score.

---

### Meta-Review · Area_Chair_pKis · 2024-12-20

**Metareview:**

The paper received five reviews with ratings of 6, 8, 6, 6, and 6. It addresses the important problem of TTA and demonstrates effective performance. Additionally, the derivation of the generalization error bound provides theoretical guarantees. The reviews are unanimously positive, and as a result, this paper is recommended for acceptance. However, the authors are encouraged to include more detailed explanations regarding the novelty, ablation studies, and sensitivity analysis in the final version of the paper.

**Additional Comments On Reviewer Discussion:**

Four reviewers engaged in discussions with the authors during rebuttal.  After the author rebuttal, two reviewers raised their review scores.

---

### Decision · Program_Chairs · 2025-01-22

Accept (Poster)